# Activation-Free Backbones for Image Recognition:
# Polynomial Alternatives within MetaFormer-Style Vision Models

**Jeffrey Wang** [1]   **Jonathan Gregory** [1]   **Grigorios G Chrysos** [1]

## Abstract

Modern vision backbones treat pointwise activations (e.g., ReLU, GELU) and exponential softmax as essential sources of nonlinearity, but we demonstrate they are not required within MetaFormer-style vision backbones. We design activation-free polynomial alternatives for three core primitives (MLPs, convolutions, and attention), where Hadamard products replace standard nonlinearities to yield polynomial functions of the input. These modules integrate seamlessly into existing architectures: instantiated within MetaFormer, a modular framework for vision backbones, our PolyNeXt models match or exceed activation-based counterparts across model scales on ImageNet classification, ADE20K semantic segmentation, and out-of-distribution robustness. We also substantially outperform prior polynomial networks at reduced computational cost, showing that polynomial variants of standard modules beat complex custom architectures. Our code is available at https://github.com/jjwang8/PolyNeXt.

## 1. Introduction

Modern vision architectures rely on pointwise activations (ReLU, GELU, SiLU) in feedforward layers and exponential functions inside softmax attention (Vaswani et al., 2017; Dosovitskiy et al., 2021). These activation functions were introduced to provide nonlinearity, enabling neural networks to approximate complex functions. However, activation functions are not the only source of nonlinearity. Polynomials offer a fundamental and well-understood alternative: by the Stone-Weierstrass theorem, any continuous function on a compact domain can be uniformly approximated by polynomials. Unlike activation functions, whose approxi-

mation properties depend on architectural depth and specific functional forms, polynomials provide a natural basis for function approximation.

The Hadamard product (elementwise multiplication) of two learned linear projections provides a simple mechanism for constructing polynomial representations. Given an input vector $x$, computing $(W_a x) * (W_b x)$ yields a second-degree polynomial in the input features, where $W_a$ and $W_b$ are learned weight matrices representing linear projections and $*$ denotes elementwise multiplication. By composing such operations across layers, networks can represent polynomials of arbitrary degree without any activation functions. This approach has theoretical appeal: the degree grows predictably with depth, and the resulting networks inherit the approximation guarantees of polynomial function spaces.

A growing body of work has explored polynomial networks that replace activations with multiplicative interactions (Chrysos et al., 2020; 2025). Part of this research has been motivated by Fully Homomorphic Encryption (FHE), an encryption paradigm where standard networks incur substantial overhead due to their non-polynomial activations (Brakerski et al., 2014). Recent architectures such as MONet (Cheng et al., 2024) and DTTN (Nie, 2025) have demonstrated that polynomial networks can approach the performance of activation-based models. However, these methods introduce custom architectures with polynomial modules tightly coupled to their specific designs. This limits their practical utility: adopting them requires committing to an entirely new architecture, their custom designs cannot easily incorporate future improvements to standard layers (e.g., efficient attention variants, better normalization), and when evaluated end-to-end, they still underperform activation-based counterparts.

We take a different approach: rather than designing a new architecture from scratch, we develop polynomial alternatives to the standard modules found in modern vision backbones. Our key insight is that by preserving the input-output interface of conventional components, polynomial modules remain compatible with orthogonal architectural improvements. We target three primitives: MLPs for channel mixing, convolutions for local spatial mixing, and self-attention for global spatial mixing. Here, *mixing* refers to operations that

---

[1]University of Wisconsin–Madison, Madison, WI, USA. Correspondence to: Jeffrey Wang <jjwang8@wisc.edu>.

*Proceedings of the 43rd International Conference on Machine Learning*, Seoul, South Korea. PMLR 306, 2026. Copyright 2026 by the author(s).

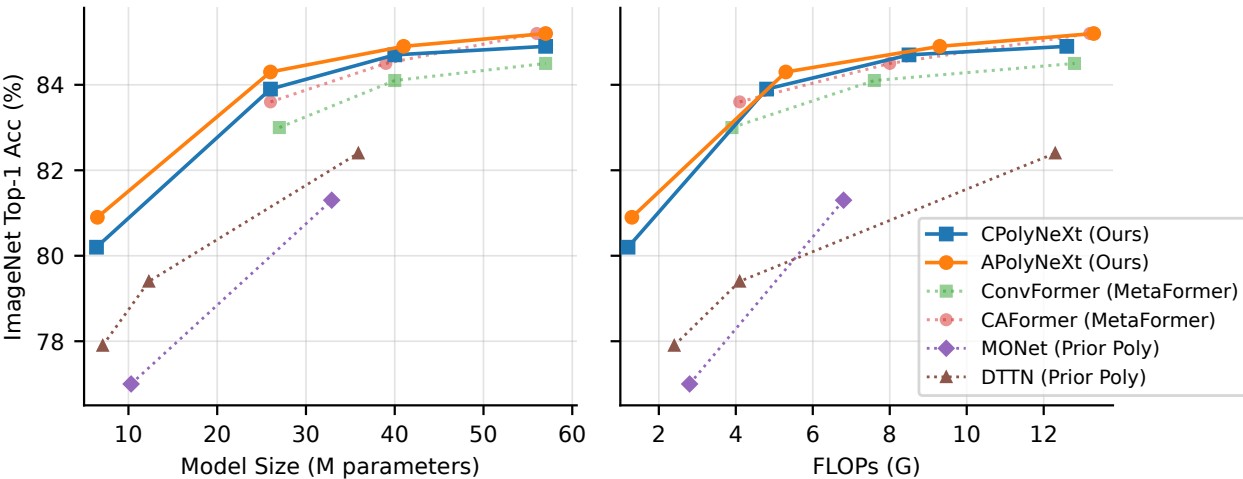

*Figure 1.* ImageNet-1K accuracy comparison. Our polynomial backbones (CPolyNeXt, APolyNeXt) consistently outperform MetaFormer instantiations (ConvFormer, CAFormer) and prior polynomial networks (MONet, DTTN) across model scales. Curves show accuracy vs. (a) parameters and (b) FLOPs.

combine information across a particular dimension, either channels (features at each spatial location) or spatial positions. These operations contain nearly all nonlinearities in contemporary backbones: MLPs apply pointwise activations between linear projections, separable convolutions interleave spatial filtering with activations, and attention uses the exponential function in softmax (Dosovitskiy et al., 2021; Liu et al., 2022; 2021). Further, by removing all activations, our work makes progress toward FHE-amenable architectures that can be used for privacy and security tasks.

The polynomial modules we introduce are:

**PolyMLP** replaces the activation in feedforward networks with a Hadamard product of two parallel linear projections:

$$\text{PolyMLP}(\boldsymbol{x}) = \boldsymbol{W}_o \left( (\boldsymbol{W}_a \boldsymbol{x}) * (\boldsymbol{W}_b \boldsymbol{x}) \right). \quad (1)$$

**PolyConv** fuses parallel convolutional branches with different receptive fields via elementwise multiplication, replacing the activation in separable convolutions.

**PolyAttn** replaces the exponential in softmax attention with a polynomial kernel, yielding an activation-free attention mechanism.

Because our changes target specific operations (activations, exponentials) rather than module structure, they remain compatible with architectural improvements such as windowed attention (Liu et al., 2021), sparse patterns (Child et al., 2019), or efficient attention variants.

We validate our modules within the MetaFormer framework (Yu et al., 2022; 2024). MetaFormer provides a modular template for vision backbones with pluggable mixing operations. Its instantiations ConvFormer and CAFormer use

exactly the primitives we target: separable convolutions, self-attention, and gated MLPs. This modular structure allows us to test each polynomial replacement independently under controlled conditions. Because these primitives are widely adopted (Shi, 2024; Fan et al., 2024; Zhu et al., 2023), we expect our polynomial replacements to transfer beyond MetaFormer to other frameworks.

Realizing the full potential of polynomial modules requires addressing stability challenges. Unlike ReLU which doesn't amplify outputs, Hadamard products of two large values produce even larger values. This multiplicative amplification compounds across layers, and we observed training instabilities in early experiments without proper controls. To address these challenges and enable stable training of networks reaching 200 layers, we introduce lightweight stabilization mechanisms: Sigmoid-Scale bounds residual contributions via a sigmoid-parameterized scalar, and multi-input skip connections (following NASNet (Zoph et al., 2018)) provide each cell with features from two preceding cells, improving gradient flow. We also find that polynomial networks benefit from a depth-over-width design: narrower layers stacked deeper outperform wider, shallower configurations at matched parameter counts (Section 3.5). Prior polynomial networks lack these mechanisms, instead using shallower, wider designs that limit polynomial degree growth.

To evaluate our polynomial modules, we instantiate them within a MetaFormer-style architecture (Yu et al., 2024), adopting its four-stage hierarchical structure while adding our stabilization mechanisms. We introduce two model variants: **CPolyNeXt**, which uses polynomial convolutions throughout, and **APolyNeXt**, which combines polynomial

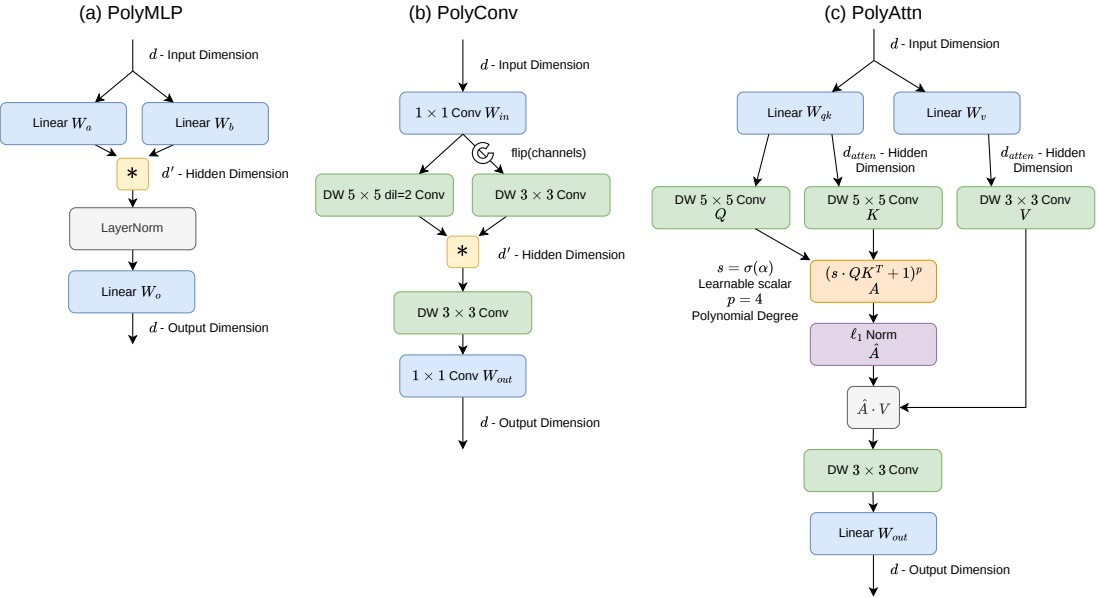

*Figure 2.* **Polynomial module primitives.** We introduce activation-free replacements for three core vision operators: (a) **PolyMLP** for channel mixing, which forms a second-degree polynomial by Hadamard-multiplying two linear projections (with LayerNorm) before an output projection; (b) **PolyConv** for convolutional spatial mixing, which replaces the activation in separable convolutions with a Hadamard fusion of a coarse (dilated) and fine depthwise branch (with channel reversal) followed by consolidation and output projection; and (c) **PolyAttn** for attention-based spatial mixing, which shares the $Q/K$ projection, injects local context via depthwise convolutions on $Q, K, V$, and replaces softmax with a polynomial attention kernel $(s \cdot QK^\top + 1)^p$ followed by $\ell_1$ normalization, where $s = \sigma(\lambda)$ is a learnable scale and $p$ is the polynomial degree. Layer-by-layer comparisons to standard MLP/Conv/Attention blocks are deferred to Appendix Fig. S4.

convolutions with polynomial attention. This design enables direct comparison against MetaFormer instantiations (ConvFormer, CAFormer) at matched scales. Our PolyNeXt backbones match or exceed these activation-based counterparts on both standard and robustness benchmarks, while outperforming prior polynomial networks at lower computational cost. Ablations confirm that reintroducing activations consistently degrades performance.

Our contributions are:

- **Polynomial alternatives to core vision primitives.** We introduce PolyMLP, PolyConv, and PolyAttn, which substitute standard nonlinearities with Hadamard products while preserving input-output interfaces for compatibility with architectural improvements.

- **Evidence that activations are unnecessary in MetaFormer-style vision backbones.** We introduce PolyNeXt, a family of polynomial vision backbones, and show that polynomial modules match or exceed activation-based counterparts across model scales, and that reintroducing activations consistently degrades performance.

- **A lightweight stabilization recipe.** We introduce Sigmoid-Scale, multi-input skip connections, and a

depth-over-width cell design, enabling stable training of polynomial networks up to 200 layers.

## 2. Related Work

**Polynomial and multiplicative networks.** Polynomial networks construct representations through multiplicative compositions rather than pointwise activations (Ivakhnenko, 1971; Shin & Ghosh, 1991). The Hadamard product of two linear projections is one such multiplicative operation used extensively in polynomial networks, providing nonlinearity without activation functions. Π-Nets formalize this construction through structured skip connections (Chrysos et al., 2020), with subsequent work addressing stability (Chrysos et al., 2023). Interested readers may refer to a recent survey for a comprehensive taxonomy of this structure (Chrysos et al., 2025). Recent architectures MONet (Cheng et al., 2024) and DTTN (Nie, 2025) propose custom polynomial architectures with specialized module designs to achieve high-accuracy classification results without relying extensively on activation functions. The Hadamard product also appears in Squeeze-and-Excitation networks (Hu et al., 2018), Gated Linear Units (Dauphin et al., 2017; Shazeer, 2020), and StyleGAN (Karras et al., 2019). Our proposed polynomial layers differ from the layers used in these previous architectures by removing activations entirely.

**MetaFormer and vision backbones.** MetaFormer abstracts vision transformer design into a modular template with pluggable mixing operations (Yu et al., 2022). The finding that even average pooling achieves competitive accuracy suggests architectural structure matters more than the specific mixer. ConvFormer and CAFormer instantiate this template with convolutions and hybrid conv-attention respectively (Yu et al., 2024). We adopt MetaFormer because its modular design enables controlled evaluation of polynomial operators against activation-based counterparts.

Related work on linear attention variants and stabilization techniques for deep networks appears in Section B.

## 3. Polynomial Modules

We present activation-free replacements for channel mixing (PolyMLP), convolutional spatial mixing (PolyConv), and attention-based spatial mixing (PolyAttn). Figure 2 summarizes the three primitives; layer-by-layer comparisons appear in Appendix Fig. S4.

### 3.1. PolyMLP: Activation-Free Channel Mixing

Standard feedforward networks apply a pointwise activation (e.g., GELU) between linear projections, and gated variants like GLU (Dauphin et al., 2017; Shazeer, 2020) add multiplicative interactions but retain an activation on one branch. PolyMLP removes activations entirely:

$$\text{PolyMLP}(\boldsymbol{x}) = \boldsymbol{W}_o\left((\boldsymbol{W}_a\boldsymbol{x}) * (\boldsymbol{W}_b\boldsymbol{x})\right), \qquad (2)$$

where $\boldsymbol{W}_a, \boldsymbol{W}_b \in \mathbb{R}^{d' \times d}$ project to an intermediate dimension $d'$, $\boldsymbol{W}_o \in \mathbb{R}^{d \times d'}$ projects back, and LayerNorm is applied after the Hadamard product. This yields a second-degree polynomial in the input (see Section A).

We set $d' = d$, corresponding to a $2\times$ expansion (smaller than the typical $4\times$), offset by increased depth (Section 3.5). For the classification head, we add an internal skip: $\text{PolyHead}(\boldsymbol{x}) = \boldsymbol{W}_o(\boldsymbol{W}_a\boldsymbol{x} + (\boldsymbol{W}_a\boldsymbol{x}) * (\boldsymbol{W}_b\boldsymbol{x}))$, preventing information from bottlenecking through the multiplicative interaction.

### 3.2. PolyConv: Polynomial Spatial Mixing via Convolution

MetaFormer's ConvFormer uses separable convolutions (depthwise spatial filtering followed by pointwise mixing) with an activation between projections (Yu et al., 2024). To replace this activation with a Hadamard product, we introduce parallel convolutional branches with different receptive fields.

The design maximizes feature diversity before multiplication. A coarse branch uses a dilated depthwise convolution ($5 \times 5$ kernel, dilation 2, covering $9 \times 9$ spatial extent) for

broad context, while a fine branch uses a standard $3 \times 3$ depthwise convolution for local detail. We apply a channel-flip (reversing channel order) to one branch before fusion to further decorrelate them:

$$\boldsymbol{h} = \boldsymbol{W}_{\text{in}}\,\boldsymbol{x}, \qquad (3)$$
$$\boldsymbol{m} = K_c(\boldsymbol{h}) * \text{flip}(K_f(\boldsymbol{h})), \qquad (4)$$
$$\boldsymbol{y} = \boldsymbol{W}_{\text{out}}\left(K(\boldsymbol{m})\right), \qquad (5)$$

where $\boldsymbol{W}_{\text{in}}$ and $\boldsymbol{W}_{\text{out}}$ are pointwise ($1 \times 1$) convolutions, $K_c$ is the dilated coarse branch, $K_f$ is the fine branch, $\text{flip}(\cdot)$ reverses channels, and $K$ is a $3 \times 3$ consolidation convolution. LayerNorm follows the block. In stage 1 (highest resolution), we use a $3 \times 3$ dilated convolution for efficiency.

Unlike prior polynomial networks that use similar structure in both branches (MONet (Cheng et al., 2024), DTTN (Nie, 2025)), PolyConv explicitly constructs heterogeneous receptive fields to produce cross-scale interaction terms. Ablations in Section 4.3 validate this design.

### 3.3. PolyAttn: Polynomial Spatial Mixing via Attention

Self-attention computes output as a weighted combination of value vectors $\boldsymbol{V}$, where weights depend on query-key similarities. Given input $\boldsymbol{X}$, standard attention projects to queries $\boldsymbol{Q} = \boldsymbol{X}\boldsymbol{W}_Q$, keys $\boldsymbol{K} = \boldsymbol{X}\boldsymbol{W}_K$, and values $\boldsymbol{V} = \boldsymbol{X}\boldsymbol{W}_V$, then computes $\text{softmax}(\boldsymbol{Q}\boldsymbol{K}^\top/\sqrt{d_k})\boldsymbol{V}$, where $d_k$ is the key dimension. The exponential in softmax provides nonlinearity.

PolyAttn replaces the exponential with a polynomial kernel. We compute unnormalized attention weights as

$$\boldsymbol{A} = \left(s \cdot \boldsymbol{Q}\boldsymbol{K}^\top + 1\right)^p, \qquad (6)$$

where $s = \sigma(\lambda)$ is a learnable per-head scalar (initialized to match the standard scale $d_k^{-1/2}$) and $p$ is the polynomial degree. We then apply $\ell_1$ normalization: $\hat{\boldsymbol{A}}_{ij} = \boldsymbol{A}_{ij}/\sum_k \boldsymbol{A}_{ik}$, and compute $\hat{\boldsymbol{A}}\boldsymbol{V}$. We use $p = 4$, which balances expressiveness and numerical stability.

Following PolyConv's structure, we augment attention with depthwise convolutions on $\boldsymbol{Q}, \boldsymbol{K}, \boldsymbol{V}$ to provide local spatial context, similar to CvT (Wu et al., 2021) and CoAtNet (Dai et al., 2021). We share the projection for queries and keys to save parameters. Because PolyAttn modifies only the attention kernel (replacing $\exp$ with a polynomial), it remains compatible with variants like windowing (Liu et al., 2021) or sparse patterns (Child et al., 2019).

### 3.4. Stabilization for Deep Polynomial Networks

Hadamard products amplify large values, and this compounds across layers. We introduce two techniques to enable stable training of networks in the hundreds of layers.

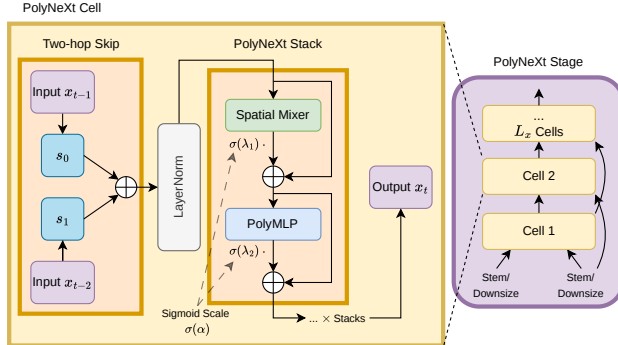

*Figure 3.* **PolyNeXt cell and stabilization.** A cell takes two inputs from the previous two cells, $x_{t-1}$ and $x_{t-2}$. In the multi-input skip module, learnable per-channel scalars $s_0$ and $s_1$ reweight $x_{t-1}$ and $x_{t-2}$ before summation and LayerNorm. The normalized state is processed by $X$ repeated PolyNeXt stacks (spatial mixer then PolyMLP). Each residual branch uses Sigmoid-Scale gating, $y = x + \sigma(\lambda) f(x)$, to bound residual updates and stabilize deep Hadamard-product networks.

**Sigmoid-Scale.** Each sublayer output is scaled by a learnable sigmoid-bounded scalar:

$$y = x + \sigma(\lambda) \cdot f(x), \qquad (7)$$

where $f(x)$ is the sublayer output and $\sigma(\lambda) \in (0,1)$ bounds the residual contribution. We initialize $\lambda$ such that $\sigma(\lambda)$ decreases with depth (see Section C.4). At inference, $\sigma(\lambda)$ can be absorbed into the preceding layer's weights, adding no activations.

**Multi-input skip connections.** We organize the network into *cells*, each containing multiple mixer-PolyMLP pairs. Following NASNet (Zoph et al., 2018), each cell receives inputs from both the previous cell ($x_{t-1}$) and the cell before that ($x_{t-2}$):

$$\tilde{x} = s_0 * x_{t-2} + s_1 * x_{t-1}, \qquad (8)$$

where $s_0, s_1 \in \mathbb{R}^C$ are learnable per-channel scalars. The combined input passes through LayerNorm before processing.

**Depth over width.** A depth-$L$ network can represent polynomials of degree $2^L$ with parameters linear in $L$, far fewer than explicitly parameterizing all degree-$2^L$ terms (Delalleau & Bengio, 2011). Stacking cells enables deeper networks than prior polynomial architectures.

### 3.5. Architecture Instantiation: PolyNeXt

We instantiate our polynomial modules within a MetaFormer-style architecture with four hierarchical stages. Spatial resolution halves at each stage transition, while channel dimension increases. An overview diagram is provided in Appendix Fig. S5.

**Variants. CPolyNeXt** uses PolyConv for spatial mixing in all stages, providing a pure convolutional polynomial

backbone. **APolyNeXt** follows CAFormer's hybrid design: PolyConv in the first two stages (high resolution, where global attention is expensive) and PolyAttn in the last two stages (low resolution, where global context is beneficial).

**Stem and downsampling.** The stem is a $7 \times 7$ convolution with stride 4. For stage 1, we initialize both skip inputs $(x_{-2}, x_{-1})$ using the stem output. For stages $s > 1$, we initialize $(x_{-2}, x_{-1})$ using the final two cell outputs from the previous stage. Downsampling between stages applies stride-2 convolutions independently to both skip paths ($x_{t-1}$ and $x_{t-2}$) before summation.

**Configuration.** Table S9 in Appendix summarizes model configurations. We refer to one mixer-PolyMLP pair as a *stack*; the number of stacks per cell increases with model size.

## 4. Experiments

We evaluate our polynomial modules on ImageNet-1K classification, comparing against MetaFormer instantiations and prior polynomial networks. Ablations isolate the contribution of each design choice, demonstrating that polynomial modules combined with our depth-over-width philosophy and stabilization techniques outperform activation-based alternatives.

### 4.1. Experimental Setup

**Dataset.** We train and evaluate on ImageNet-1K (Deng et al., 2009), the standard large-scale benchmark for image classification. The dataset contains 1.28M training images and 50K validation images across 1,000 classes. All models are trained at $224 \times 224$ resolution without additional pre-training data.

**Training recipe.** Our training configuration builds on MetaFormer (Yu et al., 2024) and MONet (Cheng et al., 2024) with modifications for polynomial networks. We use a smaller batch size (1024 vs. 4096) and increased regularization, as polynomial networks have higher capacity and benefit from stronger regularization. Complete training details, including our dropout and stochastic depth schedules, are provided in Section C.

**Baselines.** We compare against: (1) MetaFormer instantiations ConvFormer and CAFormer (Yu et al., 2024), which use the same primitives we target; (2) prior polynomial networks MONet (Cheng et al., 2024), DTTN (Nie, 2025), and StarNet (Ma et al., 2024); and (3) other strong baselines.

**Model scales.** We evaluate four model sizes: Tiny (T, $\sim$6M params), Small (S, $\sim$26M params), Base (B, $\sim$40M params), and Large (L, $\sim$60M params). Ablations are conducted at the Tiny scale for computational efficiency.

*Table 1.* **ImageNet-1K classification results.** PolyNeXt variants match or exceed comparable activation-based models across all scales while outperforming prior polynomial networks by 2–3 points. Models trained at $224 \times 224$ resolution without additional pre-training data.♦ indicates activation-free (polynomial) models. †StarNet uses element-wise multiplication but retains activations.

**(a) Conv / MLP / Polynomial Conv**

| Model | Params (M) | FLOPs (G) | Top-1 (%) |
|---|---|---|---|
| *Tiny models (<12M params)* | | | |
| MogaNet-T (Li et al., 2024) | 5.2 | 1.1 | 79.0 |
| StarNet-S4† (Ma et al., 2024) | 7.5 | 1.1 | 78.4 |
| DTTN-T♦ (Nie, 2025) | 7.1 | 2.4 | 77.9 |
| MONet-T♦ (Cheng et al., 2024) | 10 | 2.8 | 77.0 |
| **CPolyNeXt-T♦** | 6.4 | 1.2 | **80.2** |
| *Small models (∼12–30M params)* | | | |
| ConvNeXt-T (Liu et al., 2022) | 29 | 4.5 | 82.1 |
| ConvFormer-S18 (Yu et al., 2024) | 27 | 3.9 | 83.0 |
| DTTN-S♦ (Nie, 2025) | 12 | 4.1 | 79.4 |
| **CPolyNeXt-S♦** | 26 | 4.8 | **83.9** |
| *Medium models (∼30–50M params)* | | | |
| ConvNeXt-S (Liu et al., 2022) | 50 | 8.7 | 83.1 |
| MogaNet-B (Li et al., 2024) | 44 | 9.9 | 84.3 |
| InternImage-S (Wang et al., 2023) | 50 | 8.0 | 84.2 |
| UniConvNet-S (Wang & Xi, 2025) | 50 | 8.5 | 84.5 |
| ConvFormer-S36 (Yu et al., 2024) | 40 | 7.6 | 84.1 |
| MONet-S♦ (Cheng et al., 2024) | 33 | 6.8 | 81.3 |
| DTTN-B♦ (Nie, 2025) | 36 | 12.3 | 82.4 |
| **CPolyNeXt-B♦** | 40 | 8.5 | **84.7** |
| *Large models (∼50–100M params)* | | | |
| MogaNet-L (Li et al., 2024) | 83 | 15.9 | 84.7 |
| ConvNeXt-B (Liu et al., 2022) | 89 | 15.4 | 83.8 |
| ConvFormer-M36 (Yu et al., 2024) | 57 | 12.8 | 84.5 |
| **CPolyNeXt-L♦** | 57 | 12.6 | **84.9** |

**(b) Attention / Hybrid**

| Model | Params (M) | FLOPs (G) | Top-1 (%) |
|---|---|---|---|
| *Tiny models (<12M params)* | | | |
| FAN-T-Hybrid (Zhou et al., 2022) | 7.0 | 3.5 | 80.1 |
| **APolyNeXt-T♦** | 6.5 | 1.3 | **80.9** |
| *Small models (∼12–30M params)* | | | |
| CSWin-T (Dong et al., 2022) | 23 | 4.3 | 82.7 |
| UniFormer-S (Li et al., 2023) | 22 | 3.6 | 82.9 |
| BiFormer-S (Zhu et al., 2023) | 26 | 4.5 | 83.8 |
| TransNeXt-Tiny (Shi, 2024) | 28 | 5.7 | 84.0 |
| RMT-S (Fan et al., 2024) | 27 | 4.5 | 84.1 |
| CAFormer-S18 (Yu et al., 2024) | 26 | 4.1 | 83.6 |
| **APolyNeXt-S♦** | 26 | 5.3 | **84.3** |
| *Medium models (∼30–50M params)* | | | |
| CSWin-S (Dong et al., 2022) | 35 | 6.9 | 83.6 |
| UniFormer-B (Li et al., 2023) | 50 | 8.3 | 83.9 |
| TransNeXt-Small (Shi, 2024) | 50 | 10.3 | 84.7 |
| CAFormer-S36 (Yu et al., 2024) | 39 | 8.0 | 84.5 |
| **APolyNeXt-B♦** | 41 | 9.3 | **84.9** |
| *Large models (∼50–100M params)* | | | |
| CSWin-B (Dong et al., 2022) | 78 | 15.0 | 84.2 |
| MaxViT-S (Tu et al., 2022) | 69 | 11.7 | 84.5 |
| BiFormer-B (Zhu et al., 2023) | 57 | 9.8 | 84.3 |
| TransNeXt-Base (Shi, 2024) | 90 | 18.4 | 84.8 |
| RMT-B (Fan et al., 2024) | 54 | 9.7 | 85.0 |
| CAFormer-M36 (Yu et al., 2024) | 56 | 13.2 | **85.2** |
| **APolyNeXt-L♦** | 57 | 13.3 | **85.2** |

## 4.2. Main Results

Table 1 presents ImageNet-1K classification results. Our polynomial backbones consistently match or exceed MetaFormer instantiations across all scales while substantially outperforming prior polynomial networks by 2–3 percentage points at lower computational cost. These results demonstrate that activation functions are not necessary for competitive vision backbones. Compared to prior polynomial networks, the gains are substantial: CPolyNeXt-S (26M, 4.8G FLOPs, 83.9%) outperforms DTTN-B (36M, 12.3G FLOPs, 82.4%) by 1.5 points despite using 28% fewer parameters and 61% fewer FLOPs. APolyNeXt further demonstrates that polynomial attention is a viable alternative to softmax attention in vision backbones.

Beyond matching our direct baselines, polynomial modules compete favorably with architectures that introduce specialized mechanisms. On the convolutional side, CPolyNeXt-B (40M) achieves 84.7%, outperforming InternImage-S (Wang et al., 2023) (50M, 84.2%) which uses deformable convolutions, and matching UniConvNet-S (Wang & Xi, 2025) (50M, 84.5%) at 80% of the parameters. CPolyNeXt-L (57M, 84.9%) exceeds MogaNet-L (Li et al., 2024) (83M, 84.7%), which employs multi-order gating aggregation, while using only 69% of the parameters. At the largest

scale, CPolyNeXt-L also surpasses ConvFormer-B36 (Yu et al., 2024) (100M, 84.8%) while using roughly half the parameters and FLOPs. On the attention side, APolyNeXt-S (26M, 84.3%) surpasses both RMT-S (Fan et al., 2024) (27M, 84.1%), which introduces retentive self-attention, and TransNeXt-Tiny (Shi, 2024) (28M, 84.0%), which uses pixel-focused attention. At medium scale, APolyNeXt-B (41M, 84.9%) outperforms TransNeXt-Small (Shi, 2024) (50M, 84.7%) at 82% of parameters. These comparisons suggest that polynomial nonlinearity, when properly stabilized, provides sufficient expressiveness without requiring complex architectural innovations. Figure 1 visualizes these favorable scaling properties. The depth-over-width design does incur a throughput cost relative to MetaFormer at matched scale (discussed in Section 5), but our models simultaneously use substantially less peak GPU memory than both MetaFormer and prior polynomial networks, and remain faster than prior polynomial networks at matched accuracy. The full throughput-memory comparison is reported in Table S11.

## 4.3. Ablation Studies

We conduct ablations on CPolyNeXt-T and APolyNeXt-T to isolate the contribution of each design choice.

*Table 2*. **Robustness evaluation.** Polynomial networks achieve superior robustness, matching models with nearly twice the parameters on out-of-distribution benchmarks. Evaluated without fine-tuning. ♦ indicates activation-free. mCE: mean corruption error (lower is better).

| Model | Clean | IN-C↓ | IN-A | IN-R | IN-Sk |
|---|---|---|---|---|---|
| *Small models* (∼25–30M params) | | | | | |
| Swin-T | 81.3 | 62.0 | 21.6 | 41.3 | 29.1 |
| ConvNeXt-T | 82.1 | 53.2 | 24.2 | 47.2 | 33.8 |
| ConvFormer-S18 | 83.0 | 51.7 | 25.3 | 48.7 | 35.2 |
| CAFormer-S18 | 83.6 | 47.4 | 33.5 | 48.7 | 36.6 |
| **CPolyNeXt-S♦** | 83.9 | 47.9 | 35.1 | 49.4 | **37.8** |
| **APolyNeXt-S♦** | **84.3** | **45.0** | **39.6** | **49.7** | 37.5 |
| *Medium models* (∼35–50M params) | | | | | |
| Swin-S | 83.0 | 52.7 | 32.3 | 45.1 | 32.4 |
| ConvNeXt-S | 83.1 | 51.2 | 31.2 | 49.5 | 37.1 |
| MONet-S♦ | 81.3 | 49.7 | – | – | – |
| ConvFormer-S36 | 84.1 | 47.1 | 33.2 | 50.8 | 38.4 |
| CAFormer-S36 | 84.5 | 44.7 | 40.9 | 51.7 | 39.5 |
| **CPolyNeXt-B♦** | 84.7 | 44.5 | 42.8 | 52.0 | 40.0 |
| **APolyNeXt-B♦** | **84.9** | **42.7** | **46.8** | **52.8** | **41.1** |
| *Large models* (∼50–90M params) | | | | | |
| Swin-B | 83.5 | 54.4 | 35.8 | 46.6 | 32.4 |
| ConvNeXt-B | 83.8 | 46.8 | 36.7 | 51.3 | 38.2 |
| ConvFormer-M36 | 84.5 | 46.5 | 37.6 | 51.0 | 39.2 |
| CAFormer-M36 | **85.2** | 42.6 | 45.6 | 51.7 | 39.6 |
| **CPolyNeXt-L♦** | 84.9 | **42.5** | 48.3 | **54.5** | **41.8** |
| **APolyNeXt-L♦** | **85.2** | 42.9 | **49.2** | 54.0 | **41.8** |

*Table 3*. **Polynomial modules vs. activations.** Reintroducing activations degrades performance (negative Δ). CPolyNeXt-T baseline: 80.2%. APolyNeXt-T baseline: 80.9%. [†]Requires warmup and post-MLP LayerNorm. [‡]Number of heads reduced to match compute.

| Configuration | Δ Acc (%) |
|---|---|
| *CPolyNeXt-T* | |
| PolyMLP → MLP+GELU[†] | −0.1 to −0.4 |
| PolyConv → SepConv+GELU | −0.9 |
| + $\sigma(x) * x$ (act one branch) | −0.4 |
| + $\sigma(x * x)$ (act after mult) | −1.0 |
| + $\sigma(x) * \sigma(x)$ (act both) | −0.6 |
| Fine branch only (no coarse) | −0.5 |
| Hadamard → Addition | **−22.3** |
| *APolyNeXt-T* | |
| PolyMLP → MLP+GELU | −0.3 |
| PolyAttn → Std Attn[‡] | −1.3 |
| Poly kernel + $\ell_1$ → Softmax | −0.1 |
| Polynomial degree $p = 3$ | −0.3 |
| Polynomial degree $p = 5$ | −0.1 |

**Polynomial modules vs. activations.** Table 3 tests our central claim by reintroducing activations into polynomial modules. Across all modules, activations consistently degrade performance. PolyConv is most sensitive: replacing it with a standard separable convolution costs 0.9 points, and any form of activation insertion (before, after, or around the Hadamard product) hurts accuracy. For PolyAttn, the specific kernel choice matters less than the overall design where swapping our polynomial kernel for softmax costs only 0.1 points, but replacing PolyAttn entirely with standard attention drops 1.3 points, indicating that the shared $Q/K$ projection and depthwise convolutions contribute more than the kernel. Most strikingly, replacing Hadamard products with addition causes catastrophic failure (−22.3 points), confirming that multiplicative interaction is essential to the polynomial formulation.

**Stabilization and architecture.** Table 4 validates our stabilization techniques and depth-over-width design. Sigmoid-Scale is critical, but the initialization geometry, not the sigmoid nonlinearity itself, is the primary mechanism: replacing $\sigma(\lambda)$ with a free learnable scalar initialized to the same value costs only 0.5 points, while LayerScale fails even with small initialization ($10^{-6}$, −0.8 points) and standard initialization causes training collapse (−12.8 points). The smaller 0.5-point gap between the sigmoid

and free-scalar variants reflects a secondary optimization benefit: $\sigma'(\lambda) = \sigma(\lambda)(1-\sigma(\lambda))$ shrinks for deeper sublayers (smaller $\sigma(\lambda)$), tying update magnitude to current contribution size. Multi-input skip connections and pre-cell normalization each contribute meaningfully, and deeper configurations consistently outperform wider ones at matched parameter counts: 3 stacks per cell beats 1 stack by 1.5 points, validating that polynomial networks benefit from depth, which increases polynomial degree exponentially.

**Recipe sensitivity.** A natural question is whether our polynomial models depend on their custom training recipe, or whether the modules themselves drive performance. We address this with cross-recipe experiments at Small scale (Table 5), retraining ConvFormer-S18 and CAFormer-S18 in their codebase under matched speedups (`torch.compile`, AMP bf16) to within 0.1 of published numbers (82.9% and 83.5%). Optimizer choice is where model-specificity lives, but regularization transfers in both directions: ConvFormer under our regularization is essentially unchanged (+0.1), and CAFormer under our full recipe drops only 0.3, while ConvFormer *collapses* under our optimizer. Both CPolyNeXt-S (−0.2) and APolyNeXt-S (−0.4) are nearly unaffected by ConvFormer's optimizer, comparable to what MetaFormer models themselves experience under recipe swaps; the polynomial modules are no more recipe-sensitive than standard architectures.

**Module drop-in.** To further isolate module-level generality, we replace the spatial mixers in CAFormer-S18 with PolyConv and PolyAttn while keeping CAFormer's architecture and training recipe and matching FLOPs/parameters. Both substitutions improve ImageNet accuracy by +0.1 to +0.3 points, indicating that the polynomial modules transfer

*Table 4.* **Stabilization and architecture ablations.** Deeper configurations outperform wider ones at matched parameters. CPolyNeXt-T baseline: 80.2%. [†]Training fails to converge until later epochs. [‡]Learnable scalar initialized to $\sigma(\lambda)$ but without the sigmoid.

| Configuration | $\Delta$ Acc (%) |
|---|---|
| *Sigmoid-Scale ablations* | |
| $\rightarrow$ Learnable scalar (no $\sigma$)[‡] | −0.5 |
| $\rightarrow$ LayerScale (init=$10^{-6}$) | −0.8 |
| $\rightarrow$ LayerScale (init=1.0)[†] | **−12.8** |
| *Skip connection ablations* | |
| Remove multi-input skip | −0.6 |
| Remove norm before cell | −0.4 |
| *Depth vs. width (matched params)* | |
| Wider (2 stacks/cell) | −0.7 |
| Wider (1 stack/cell) | −1.5 |

*Table 5.* **Cross-recipe analysis at Small scale.** $\Delta$ accuracy from each model's own baseline when trained with the other family's optimizer and/or regularization. Optimizer choice is model-specific; regularization transfers in both directions.

| Configuration | $\Delta$ Acc (%) |
|---|---|
| *Ours w/ ConvFormer settings* (CPolyNeXt-S 83.9%, APolyNeXt-S 84.3%) | |
| CPolyNeXt-S w/ ConvFormer optimizer | −0.2 |
| APolyNeXt-S w/ ConvFormer optimizer | −0.4 |
| *ConvFormer-S18 w/ our settings* (baseline 82.9%) | |
| w/ our optimizer (w/ or w/o our reg.) | **collapse (∼6%)** |
| w/ our regularization | +0.1 |
| *CAFormer-S18 w/ our settings* (baseline 83.5%) | |
| w/ our optimizer | −0.4 |
| w/ our regularization | −0.3 |
| w/ our full recipe | −0.3 |

beyond the PolyNeXt architecture itself.

## 4.4. Robustness Evaluation

Following MetaFormer (Yu et al., 2024), we evaluate robustness on ImageNet-C (Hendrycks & Dietterich, 2019) (common corruptions such as noise and blur), ImageNet-A (Hendrycks et al., 2021b) (natural adversarial examples), ImageNet-R (Hendrycks et al., 2021a) (renditions such as paintings and sculptures), and ImageNet-Sketch (Wang et al., 2019) (sketch representations) without fine-tuning (Table 2). Our polynomial backbones demonstrate superior robustness across all MetaFormer instantiations at comparable scales, with APolyNeXt showing particular gains on adversarial and out-of-distribution data. Notably, our medium-scale convolutional model CPolyNeXt-B (40M parameters) surpasses ConvFormer-B36 (100M) on ImageNet-A (42.8 vs. 40.1), ImageNet-R (52.0 vs. 51.1), and ImageNet-Sketch (40.0 vs. 39.5), while CPolyNeXt-L (57M) matches or exceeds CAFormer-B36 (99M) on ImageNet-C (42.5 vs. 42.6)

*Table 6.* **ADE20K semantic segmentation.** UperNet with 160K iterations under the ConvNeXt recipe. Polynomial backbones surpass both MetaFormer baselines at matched parameters.

| Model | Params (M) | MACs (G) | mIoU |
|---|---|---|---|
| ConvFormer-S18 (Yu et al., 2024) | 54 | 925 | 48.6 |
| CAFormer-S18 (Yu et al., 2024) | 54 | 1024 | 48.9 |
| **CPolyNeXt-S**♦ | 54 | 942 | **50.6** |
| **APolyNeXt-S**♦ | 55 | 1121 | 49.9 |

and ImageNet-R (54.5 vs. 53.9) at nearly half the parameters. These results suggest that polynomial networks offer favorable robustness-efficiency trade-offs beyond clean accuracy.

## 4.5. Dense Prediction on ADE20K

To test transfer beyond classification, we evaluate semantic segmentation on ADE20K (Zhou et al., 2019) using UperNet (Xiao et al., 2018) following the ConvNeXt (Liu et al., 2022) training recipe (160K iterations, AdamW with lr=$10^{-4}$ and weight decay 0.05), with PolyNeXt-specific parameter groups (no weight decay on Sigmoid-Scale, multi-input skip, and normalization parameters) and no further tuning. MetaFormer baseline numbers are from Yu et al. (2024). As shown in Table 6, CPolyNeXt-S surpasses both ConvFormer-S18 (+2.0 mIoU) and CAFormer-S18 (+1.7 mIoU) at matched parameters, and matches ConvFormer-S36 (50.7 mIoU at 67M params, 1003G MACs) with 54M params and 941G MACs. Segmentation gains thus exceed our classification margins, suggesting polynomial backbones transfer particularly well to dense prediction. APolyNeXt-S required per-cell gradient clipping at norm 5; the convolutional variant needed no modifications.

## 4.6. Towards FHE: Fully Polynomial Variants

While our main models remove all pointwise activations, LayerNorm remains a barrier to FHE-compatible inference because it requires division and square-root on encrypted values. To address this, we train *fully* polynomial variants in which every LayerNorm is replaced by a polynomial-compatible BatchNorm: it reduces over batch and channels $(B, C)$ per spatial position with a factorized affine and running statistics at inference. The PolyAttn $\ell_1$ normalization is replaced analogously with a running row-sum estimate (full derivation in Section C.5). The entire inference pass thus involves only additions and multiplications.

As shown in Table 7, our CPolyNeXt-S BN variant reaches 82.7% on ImageNet, a 5.0-point improvement over the best prior fully polynomial result (MONet-T BN, 77.7%). Notably, CPolyNeXt-S BN (82.7%, 26M) surpasses ConvNeXt-T (82.1%, 29M), a strong activation-based baseline, at fewer parameters, and approaches

*Table 7.* **Fully polynomial variants.** Replacing LayerNorm with polynomial-compatible BatchNorm makes the entire inference pass additions and multiplications only. CPolyNeXt-S BN at 82.7% surpasses ConvNeXt-T (82.1%) and is +5.0 over the best prior fully polynomial model.

| Model | Params | FLOPs | LN ver. | Poly BN | Δ |
|---|---|---|---|---|---|
| *Prior polynomial networks* | | | | | |
| MONet-T (BN)♦ | 10.3M | 2.8G | 77.0 | 77.7 | +0.7 |
| DTTN-S (no norm)♦ | 12.3M | 4.1G | 79.4 | 77.2 | −2.2 |
| *Ours* | | | | | |
| **CPolyNeXt-T**♦ | 6.4M | 1.2G | 80.2 | 78.3 | −1.9 |
| **APolyNeXt-T**♦ | 6.5M | 1.3G | 80.9 | 78.0 | −2.9 |
| **CPolyNeXt-S**♦ | 26M | 4.8G | 83.9 | **82.7** | −1.2 |

ConvFormer-S18 (83.0%) within 0.3 points. The LN→BN gap is larger for the attention variant (APolyNeXt-T: −2.9 vs. CPolyNeXt-T: −1.9), reflecting the additional approximation in the $\ell_1$ attention normalization, and shrinks with scale for the convolutional variant (CPolyNeXt-S: −1.2).

## 5. Discussion

**From capacity to trainability.** Our results suggest that the primary barrier to competitive polynomial networks was not representational capacity but optimization stability. Polynomial degree grows exponentially with depth while parameters grow only linearly, making deep polynomial networks highly expressive. This expressiveness manifests empirically as a need for stronger regularization (smaller batch sizes, higher stochastic depth). However, depth also amplifies instability: multiplying two large values produces even larger values, and this compounds across layers. Prior polynomial networks avoided this instability through shallower, wider designs, sacrificing the capacity benefits of depth. Our stabilization techniques address this through controlled residual magnitudes (Sigmoid-Scale), improved gradient flow (multi-input skip connections), and careful normalization placement. With these in place, we can train networks just under 200 layers, enabling polynomial modules to match or exceed activation-based counterparts (Table 1).

**Why do activations hurt?** Adding activations to polynomial modules might be expected to add expressiveness, but within our depth-over-width architecture the opposite occurs. The block $(\boldsymbol{W}_a \boldsymbol{x}) * (\boldsymbol{W}_b \boldsymbol{x})$ exhibits a mutual gradient coupling: in the backward pass, the gradient flowing into $\boldsymbol{W}_a$ is scaled by $\boldsymbol{W}_b \boldsymbol{x}$ and vice versa, so each branch learns through its sibling's output. Adding GELU breaks this coupling, since its near-zero derivative for negative inputs creates dead zones where gradient flow is suppressed. This explains the ordering in Table 3, where $\sigma$ denotes GELU: applying it to *one branch* ($\sigma(\boldsymbol{W}_a \boldsymbol{x}) * (\boldsymbol{W}_b \boldsymbol{x})$, $\Delta = -0.4$) partially blocks one gradient path; applying it

to *both branches* ($\sigma(\boldsymbol{W}_a \boldsymbol{x}) * \sigma(\boldsymbol{W}_b \boldsymbol{x})$, $\Delta = -0.6$) blocks both paths independently; and applying it *after the product* ($\sigma((\boldsymbol{W}_a \boldsymbol{x}) * (\boldsymbol{W}_b \boldsymbol{x}))$, $\Delta = -1.0$) is worst because a single gate blocks gradients to *both* projections simultaneously whenever the product is negative. Replacing multiplication with addition ($\Delta = -22.3$) removes the mutual gradient coupling entirely, confirming that this coupling is the essential source of expressive nonlinearity in our polynomial modules.

**Limitations.** Our training recipe does not transfer directly to standard pipelines: polynomial networks require smaller batch sizes, progressive dropout schedules, and careful initialization to train stably. The depth-over-width design incurs throughput overhead compared to shallower, wider architectures, even at matched FLOPs. Hadamard products are sensitive to learning rate due to multiplicative amplification. Finally, while our fully polynomial BN variants (Section 4.6) show that LayerNorm is not an absolute barrier to activation- and normalization-free inference, end-to-end FHE deployment remains future work.

## 6. Conclusion

We introduced polynomial alternatives for the core modules in modern vision backbones: PolyMLP for channel mixing, PolyConv for convolutional spatial mixing, and PolyAttn for attention-based spatial mixing. Each module substitutes pointwise activations or exponential operations with Hadamard products while preserving input-output interfaces. Combined with lightweight stabilization techniques and a depth-over-width design, these modules match or exceed activation-based counterparts across model scales on ImageNet classification and out-of-distribution robustness, and transfer favorably to ADE20K semantic segmentation, where our CPolyNeXt-S surpasses both ConvFormer-S18 and CAFormer-S18 at matched parameters (Section 4.5). We further show that the activation-free paradigm can be extended to normalization: by replacing LayerNorm with a polynomial-compatible BatchNorm, our *fully* polynomial variants reach 82.7% on ImageNet, a 5.0-point advance over the best prior fully polynomial model and a step toward FHE-compatible inference (Section 4.6).

Beyond the empirical results, our work suggests that activation functions, at least for image recognition within modern vision architectures, are not a representational necessity but rather one solution to the optimization challenges of deep networks. Once training stability is addressed through other means, polynomial interactions can serve as effective sources of nonlinearity. Future directions include characterizing the optimization dynamics of polynomial networks, extending to domains such as language modeling where softmax attention is ubiquitous, and FHE deployment.

## Acknowledgments

We thank our anonymous reviewers for their thoughtful and constructive feedback, which substantially improved this work. This research was performed using the compute resources and assistance of the UW–Madison Center for High Throughput Computing (CHTC) (Center for High Throughput Computing, 2006). We also thank Zulip for the support of our online communication.

## Impact Statement

This paper presents fundamental research on vision backbone architectures, investigating whether activation functions are necessary components of modern neural networks. As general-purpose vision backbones, our models inherit the dual-use considerations common to computer vision research. One potential positive impact is enabling privacy-preserving inference: by removing activation functions, our architectures take a step toward models compatible with Fully Homomorphic Encryption, though significant barriers remain. We note, however, that our activation-free design does not by itself imply safer, fairer, or more privacy-preserving deployment, and those properties require task-specific validation, which is left for future work. We do not foresee negative societal consequences unique to activation-free architectures beyond those inherent to advancing vision model capabilities generally.

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

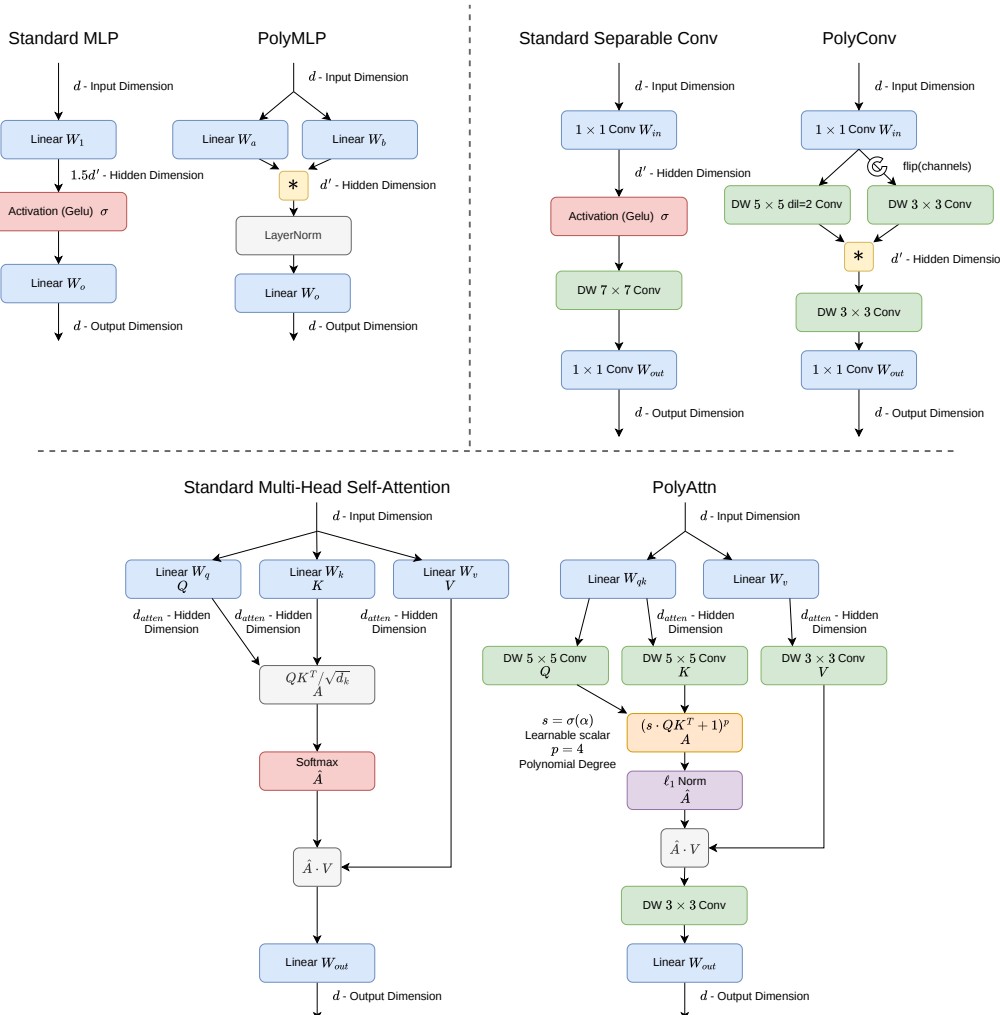

*Figure S4.* **Layer by layer comparisons to standard vision primitives.** Top left: a standard MLP applies a pointwise activation (GELU) between linear projections, while **PolyMLP** replaces the activation with a Hadamard product of two parallel projections followed by LayerNorm and an output projection. Top right: a standard separable convolution block applies an activation between input and output projections, while **PolyConv** replaces the activation with a Hadamard fusion of two depthwise branches with different receptive fields (a coarse dilated branch and a fine branch with channel reversal), followed by a consolidation convolution and output projection. Bottom: standard multi-head self-attention uses separate $W_q, W_k, W_v$ and softmax normalization of $QK^\top/\sqrt{d_k}$, while **PolyAttn** shares the $Q/K$ projection, applies depthwise convolutions to $Q, K, V$, and replaces softmax with a polynomial kernel $(s \cdot QK^\top + 1)^p$ followed by $\ell_1$ normalization, where $s = \sigma(\alpha)$ is a learnable scale and $p$ is the polynomial degree.

## A. PolyMLP Computes a Second-Degree Polynomial

We show that PolyMLP computes a second-degree polynomial in the input features. For simplicity, we omit the LayerNorm, which does not affect the polynomial degree.

**Proposition A.1.** *Let $\boldsymbol{x} \in \mathbb{R}^d$ be the input, and let $\boldsymbol{W}_a, \boldsymbol{W}_b \in \mathbb{R}^{d' \times d}$ and $\boldsymbol{W}_o \in \mathbb{R}^{d \times d'}$ be weight matrices. The PolyMLP output*

$$PolyMLP(\boldsymbol{x}) = \boldsymbol{W}_o\left((\boldsymbol{W}_a\boldsymbol{x}) * (\boldsymbol{W}_b\boldsymbol{x})\right) \tag{9}$$

*is a second-degree polynomial in the entries of $\boldsymbol{x}$.*

*Proof.* Let $\boldsymbol{a} = \boldsymbol{W}_a \boldsymbol{x} \in \mathbb{R}^{d'}$ and $\boldsymbol{b} = \boldsymbol{W}_b \boldsymbol{x} \in \mathbb{R}^{d'}$. Each entry of $\boldsymbol{a}$ is a linear combination of the entries of $\boldsymbol{x}$:

$$a_i = \sum_{j=1}^{d} [\boldsymbol{W}_a]_{ij} x_j, \tag{10}$$

and similarly for $\boldsymbol{b}$. The Hadamard product yields:

$$[\boldsymbol{a} * \boldsymbol{b}]_i = a_i \cdot b_i = \left( \sum_{j=1}^{d} [\boldsymbol{W}_a]_{ij} x_j \right) \left( \sum_{k=1}^{d} [\boldsymbol{W}_b]_{ik} x_k \right). \tag{11}$$

Expanding this product gives a sum of terms of the form $[\boldsymbol{W}_a]_{ij}[\boldsymbol{W}_b]_{ik} x_j x_k$, which are degree-2 monomials in the entries of $\boldsymbol{x}$. The final output projection $\boldsymbol{W}_o$ takes linear combinations of these degree-2 terms, preserving the polynomial degree. Therefore, each entry of $\text{PolyMLP}(\boldsymbol{x})$ is a polynomial of degree at most 2 in the entries of $\boldsymbol{x}$. $\square$

This proof follows analogously from the analysis in MONet (Cheng et al., 2024), adapted to our simpler symmetric structure.

## B. Extended Related Work

**Attention beyond exponential softmax.** Alternatives to exponential softmax include Performers (Choromanski et al., 2021), Linear Transformer (Katharopoulos et al., 2020), cosFormer (Qin et al., 2022), and BASED (Arora et al., 2024). These methods primarily target computational efficiency by avoiding the quadratic cost of full attention. Our PolyAttn has a different motivation: demonstrating that the exponential function is not essential for competitive accuracy. We replace the exponential with a polynomial kernel and apply $\ell_1$ normalization, yielding an activation-free attention mechanism. Because this change targets only the attention kernel computation, it remains compatible with structural improvements like windowing (Liu et al., 2021) or sparse patterns.

**Stabilizing deep networks.** Residual connections (He et al., 2016) enable training of very deep networks by providing gradient shortcuts. LayerScale (Touvron et al., 2021b) introduces learnable per-channel scaling of residual contributions, improving training stability for vision transformers. Multi-hop skip connections, as in DenseNet (Huang et al., 2017) and NASNet (Zoph et al., 2018), provide additional gradient pathways by connecting layers to multiple predecessors.

Polynomial networks present particular stability challenges because Hadamard products can amplify activation magnitudes: unlike ReLU, which does not magnify outputs, the multiplication of two large values in a Hadamard product produces an even larger value. Prior polynomial networks address this through regularization (Chrysos et al., 2023) or architectural constraints (Cheng et al., 2024). We introduce Sigmoid-Scale, which bounds residual contributions through a sigmoid-parameterized learnable scalar, and adopt multi-input skip connections following NASNet. This lightweight approach preserves the core polynomial operations while enabling training at around 200 layers.

## C. Training Details

### C.1. Implementation

All experiments are conducted using PyTorch with `torch.compile` and automatic mixed precision (AMP) with bfloat16. Training is performed on $2\times$ NVIDIA RTX 6000 Ada Generation GPUs.

### C.2. Hyperparameters

Table S8 provides complete training hyperparameters for all model sizes.

### C.3. Comparison with MetaFormer Training Recipe

Our training recipe differs from MetaFormer (Yu et al., 2024) in several ways, with some modifications following MONet (Cheng et al., 2024). The key insight is that polynomial networks have higher capacity than their activation-based counterparts and therefore benefit from stronger regularization and modified optimization:

- **Batch size:** We use 1024 instead of 4096. We found that polynomial networks benefit from smaller batch sizes, possibly due to the different optimization landscape induced by Hadamard products.

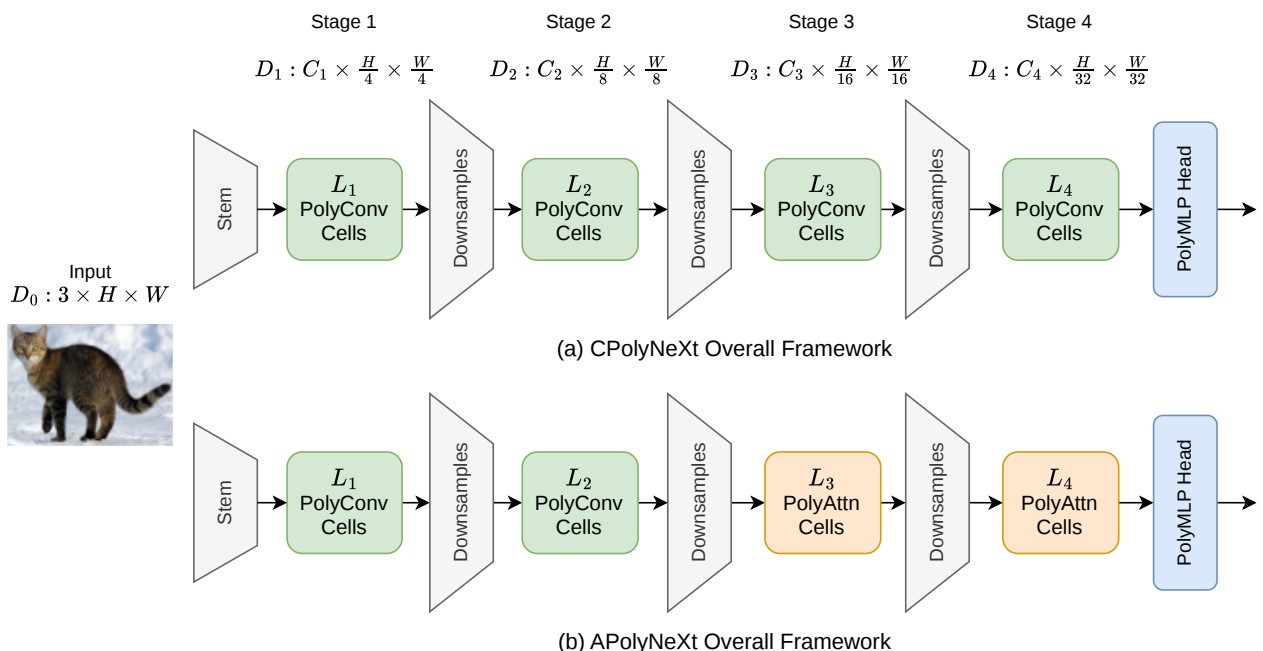

*Figure S5.* **Overall PolyNeXt architecture.** PolyNeXt follows a four-stage hierarchical design with decreasing spatial resolution and increasing channel width. The input tensor is $D_0 : 3 \times H \times W$. The stem is a $7 \times 7$ convolution with stride 4; downsampling modules use stride-2 convolutions (see Section 3.5 for details). After the stem, stage $s \in \{1, 2, 3, 4\}$ operates on $D_s : C_s \times \frac{H}{2^{s+1}} \times \frac{W}{2^{s+1}}$ and contains $L_s$ cells. Each cell consists of multiple stacks (spatial mixer plus PolyMLP) connected via multi-input skip connections, as detailed in Figure 3. CPolyNeXt uses PolyConv cells in all stages, while APolyNeXt uses PolyConv in stages 1–2 and PolyAttn in stages 3–4.

- **Optimizer:** We use AdamW for CPolyNeXt and LAMB for APolyNeXt (the attention variant), following MetaFormer's finding that attention models benefit from LAMB.

- **Learning rate and weight decay:** We use 0.004/0.002 learning rate for CPolyNeXt/APolyNeXt and 0.01 weight decay, adjusted for the smaller batch size.

- **Stochastic depth:** Our stochastic depth rates are set 0.05 lower than the corresponding MetaFormer model one size larger (e.g., our 26M model uses the rate from MetaFormer's 40M model minus 0.05). This increased regularization reflects the higher capacity of polynomial networks.

- **Dropout schedule:** Following NASNet's (Zoph et al., 2018) use of progressive regularization, we linearly increase dropout over training epochs rather than using constant dropout. We found this prevents numerical instabilities (NaN losses) that occur with constant dropout in deep polynomial networks.

- **Warmup:** CPolyNeXt-T trains stably without warmup; larger CPolyNeXt models use 5 warmup epochs to prevent early training instability. APolyNeXt models require no warmup due to the stabilizing effect of Sigmoid-Scale.

- **EMA:** We apply exponential moving average (EMA) of model weights for the last 200 epochs, following MONet.

- **Weight initialization:** We use Kaiming initialization with gain $\sqrt{2}$ for PolyConv, PolyMLP, and downsampling layer weights.

## C.4. Sigmoid-Scale Initialization

Each cell maintains a vector of Sigmoid-Scale parameters $\boldsymbol{\lambda}$, one per sublayer (mixer and PolyMLP alternating). For a cell with $S$ stacks (each stack containing one mixer and one PolyMLP), the vector has length $2S_{\max}$, where $S_{\max}$ is the maximum number of stacks in any cell of the network.

*Table S8.* **Training hyperparameters for PolyNeXt models.** All models are trained on ImageNet-1K for 300 epochs at $224 \times 224$ resolution. RandAugment settings follow MetaFormer.

**(a) CPolyNeXt**

| Hyperparameter | T | S | B | L |
|---|---|---|---|---|
| Optimizer | | AdamW | | |
| Batch size | | 1024 | | |
| Learning rate | | 0.004 | | |
| Weight decay | | 0.01 | | |
| LR schedule | | Cosine decay | | |
| Warmup epochs | 0 | 5 | 5 | 5 |
| Warmup LR multiplier | | 0.1 | | |
| RandAugment | | 9/0.5 | | |
| Mixup $\alpha$ | | 0.8 | | |
| CutMix $\alpha$ | | 1.0 | | |
| Random erasing prob. | | 0.25 | | |
| Label smoothing | | 0.1 | | |
| Stochastic depth | 0.0 | 0.20 | 0.30 | 0.50 |
| Dropout schedule | | Linear increase | | |
| Final dropout | 0.0 | 0.4 | 0.6 | 0.6 |

**(b) APolyNeXt**

| Hyperparameter | T | S | B | L |
|---|---|---|---|---|
| Optimizer | | LAMB | | |
| Batch size | | 1024 | | |
| Learning rate | | 0.002 | | |
| Weight decay | | 0.01 | | |
| LR schedule | | Cosine decay | | |
| Warmup epochs | | 0 | | |
| Warmup LR multiplier | | 0.1 | | |
| RandAugment | | 9/0.5 | | |
| Mixup $\alpha$ | | 0.8 | | |
| CutMix $\alpha$ | | 1.0 | | |
| Random erasing prob. | | 0.25 | | |
| Label smoothing | | 0.1 | | |
| Stochastic depth | 0.03 | 0.25 | 0.40 | 0.55 |
| Dropout schedule | | Linear increase | | |
| Final dropout | 0.1 | 0.4 | 0.6 | 0.7 |

For Small and Base models, we initialize:

$$\lambda_i = -\frac{i}{2}, \quad \text{for } i \in \{0, 1, \ldots, 2S_{\max} - 1\}. \tag{12}$$

This yields sigmoid values that decrease geometrically with depth within each cell: $\sigma(\lambda_0) = 0.50$, $\sigma(\lambda_1) \approx 0.38$, $\sigma(\lambda_2) \approx 0.27$, and so on.

For the Large model, deeper networks benefit from smaller initial scales to reduce gradient noise during early training, so we subtract an additional $0.5$ from all logits:

$$\lambda_i = -\frac{i}{2} - 0.5, \quad \text{for } i \in \{0, 1, \ldots, 2S_{\max} - 1\}. \tag{13}$$

The Tiny model uses a probability-based initialization with shared scales between adjacent blocks, which we found empirically to improve stability at the smallest scale.

### C.5. Polynomial-Compatible Normalization

We detail the normalization variants used in our fully polynomial models (Section 4.6). The key constraint is that every operation must be expressible as a polynomial in the input at inference time, since division and square root are expensive under Fully Homomorphic Encryption (FHE). Standard BatchNorm satisfies this because the running statistics become constants at inference, reducing the whole operation to a per-channel affine map. Standard LayerNorm does not: its statistics are computed per-sample, so the divisor depends on the input. Our variants combine LayerNorm-style channel reduction (which preserves the inductive bias of modern vision backbones) with BatchNorm-style running statistics (which makes the operation polynomial at inference).

**Spatial normalization: standard BatchNorm vs. ours.** Let $x \in \mathbb{R}^{B \times C \times H \times W}$. Standard `nn.BatchNorm2d` reduces over the batch and spatial axes $(B, H, W)$, producing per-channel statistics with per-channel affine parameters $\boldsymbol{\gamma}_c, \boldsymbol{\beta}_c \in \mathbb{R}^C$:

$$\mu_c = \tfrac{1}{BHW} \sum_{b,h,w} x_{bchw}, \tag{14}$$

$$\sigma_c^2 = \tfrac{1}{BHW} \sum_{b,h,w} (x_{bchw} - \mu_c)^2, \tag{15}$$

$$y_{bchw} = \gamma_c \cdot \frac{x_{bchw} - \mu_c}{\sqrt{\sigma_c^2 + \epsilon}} + \beta_c. \tag{16}$$

Our variant reduces over the batch and channel axes $(B, C)$ instead, producing per-position statistics with a *factorized* affine. In its general form this affine has a per-channel scale and bias $\boldsymbol{\gamma}_c, \boldsymbol{\beta}_c \in \mathbb{R}^C$ and a per-position scale and bias $\boldsymbol{\gamma}_{hw}, \boldsymbol{\beta}_{hw} \in \mathbb{R}^{H \times W}$:

$$\mu_{hw} = \frac{1}{BC} \sum_{b,c} x_{bchw}, \tag{17}$$

$$\sigma_{hw}^2 = \frac{1}{BC} \sum_{b,c} (x_{bchw} - \mu_{hw})^2, \tag{18}$$

$$y_{bchw} = (\gamma_c \cdot \gamma_{hw}) \cdot \frac{x_{bchw} - \mu_{hw}}{\sqrt{\sigma_{hw}^2 + \epsilon}} + (\beta_c + \beta_{hw}). \tag{19}$$

The factorization keeps the affine parameter count at $\mathcal{O}(C + HW)$ rather than $\mathcal{O}(CHW)$. In our models we remove the per-channel bias $\boldsymbol{\beta}_c$, so the realized affine uses only the factorized scale $\gamma_c \cdot \gamma_{hw}$ and the per-position bias $\beta_{hw}$; we keep $\boldsymbol{\beta}_c$ in the general form since it applies directly to architectures that use normalization biases. At inference, $\mu_{hw}$ and $\sigma_{hw}^2$ are replaced by running estimates and become fixed constants. The operation then collapses to a per-position affine transformation

$$y_{bchw} = A_{chw} \, x_{bchw} + B_{chw}, \tag{20}$$

with $A_{chw} = (\gamma_c \cdot \gamma_{hw})/\sqrt{\sigma_{hw}^2 + \epsilon}$ and $B_{chw} = (\beta_c + \beta_{hw}) - A_{chw}\mu_{hw}$, which is a degree-1 polynomial in $\boldsymbol{x}$ and therefore FHE-compatible.

In PyTorch, the difference from `nn.BatchNorm2d` is two lines (the reduction axes) plus the factorized affine:

```
# Standard nn.BatchNorm2d: reduce over (B,H,W); gamma_c, beta_c in R^C
mu  = x.mean(dim=(0, 2, 3), keepdim=True)
var = x.var (dim=(0, 2, 3), keepdim=True, unbiased=False)
y = gamma_c * (x - mu) / (var + eps).sqrt() + beta_c

# Ours: reduce over (B,C); factorized affine
mu  = x.mean(dim=(0, 1), keepdim=True)
var = x.var (dim=(0, 1), keepdim=True, unbiased=False)
y = (gamma_c * gamma_hw) * (x - mu) / (var + eps).sqrt() \
    + (beta_c + beta_hw)
# At inference, mu and var are replaced by running buffers
# and the operation reduces to a fixed affine map.
```

**PolyAttn $\ell_1$ normalization.** Standard PolyAttn divides each row of the unnormalized attention matrix $\boldsymbol{A} \in \mathbb{R}^{B \times H \times N \times N}$ by its own row sum: $\hat{A}_{bhij} = A_{bhij} / \sum_k A_{bhik}$. This division depends on the current input and is not FHE-compatible. We replace the per-sample row sum with a running estimate, averaged over the batch during training:

$$R_{hi} = \mathbb{E}_b[\textstyle\sum_k A_{bhik}], \qquad \hat{A}_{bhij} = \gamma_{hi} \cdot \frac{A_{bhij}}{R_{hi} + \epsilon}, \tag{21}$$

with a multiplicative learnable affine $\boldsymbol{\gamma} \in \mathbb{R}^{H \times N}$ (one per head and query position). There is no additive bias. The unnormalized PolyAttn weights are already non-negative by construction, so no absolute value is required. At inference, $R_{hi}$ is a fixed constant and the whole attention computation $\hat{\boldsymbol{A}}\boldsymbol{V}$ is polynomial in the input. The cost is approximation: per-sample normalization is replaced by a single distributional estimate, which is the primary source of the gap reported in Table 7 for APolyNeXt-T.

**Architectural change in APolyNeXt BN.** The standard PolyAttn block applies the output projection $\boldsymbol{W}_{\text{out}}$ directly to $\hat{\boldsymbol{A}}\boldsymbol{V}$ (after the depthwise convolution). In the fully polynomial APolyNeXt variant, we insert an additional copy of our spatial normalization (with reduction over $(B, C)$, factorized affine, running statistics) immediately before $\boldsymbol{W}_{\text{out}}$. This is the only block-level architectural change in the BN variants beyond replacing each LayerNorm with our polynomial-compatible spatial normalization.

**Training.** The BN variants use a unified optimization recipe across both backbone families. We train with AdamW (replacing LAMB for the attention variant), learning rate $10^{-3}$, weight decay 0.05, and a per-GPU batch size of 256, with multi-GPU training and gradient accumulation bringing the effective batch size back to 1024. The Tiny BN variants (CPolyNeXt-T BN and APolyNeXt-T BN) are trained without dropout or stochastic depth; CPolyNeXt-S BN retains the same dropout and

*Table S9.* PolyNeXt architecture configurations. Each stage $s$ operates at resolution $\frac{H}{2^{s+1}} \times \frac{W}{2^{s+1}}$. Each cell contains multiple *stacks* (one stack = one mixer + one PolyMLP). PolyMLP hidden dimension is $2 \times C_s$ in stages 1–2 and $1.75 \times C_s$ in stages 3–4. PolyConv uses hidden dimension $1 \times C_s$ in stages 1–2 and $0.75 \times C_s$ in stages 3–4; APolyNeXt attention is configured to match this compute budget (head dimension 32, $\lceil C_s/64 \rceil$ heads per stage).

|  | PolyNeXt-T | PolyNeXt-S | PolyNeXt-B | PolyNeXt-L |
|---|---|---|---|---|
| Channels $(C_1, C_2, C_3, C_4)$ | (48, 96, 192, 288) | (72, 144, 288, 432) | (84, 168, 336, 504) | (96, 192, 384, 576) |
| Cells $(L_1, L_2, L_3, L_4)$ | (2, 2, 6, 2) | (3, 3, 8, 3) | (3, 5, 10, 3) | (3, 6, 12, 3) |
| Stacks per cell | (3, 3, 3, 3) | (3, 4, 4, 4) | (4, 4, 4, 4) | (4, 4, 4, 4) |
| *Spatial Mixer:* | | | | |
| CPolyNeXt | | $T$ = (PolyConv, PolyConv, PolyConv, PolyConv) | | |
| APolyNeXt | | $T$ = (PolyConv, PolyConv, PolyAttn, PolyAttn) | | |

*Table S10.* **Low-resolution setup (CPolyNeXt-LR).** Models are trained from scratch at native resolution. Unless specified below, we follow the **CPolyNeXt** ImageNet-1K recipe in Table S8 and reuse the same components as CPolyNeXt-T in Table S9.

| Hyperparameter | CIFAR-10 | SVHN | Tiny ImageNet |
|---|---|---|---|
| *Architecture (shared across datasets)* | | | |
| Stages | | 3 | |
| Cells per stage $(L_1, L_2, L_3)$ | | (2, 3, 3) | |
| Channels $(C_1, C_2, C_3)$ | | (72, 144, 288) | |
| Stacks per cell | | (3, 3, 3) | |
| Params (M) | | 5.5 | |
| Other components | | Same as CPolyNeXt-T | |
| *Optimization and augmentation* | | | |
| Input resolution | $32^2$ | $32^2$ | $64^2$ |
| Optimizer | | AdamW | |
| LR schedule | | Cosine decay | |
| Batch size | | 96 | |
| Learning rate | | 0.001 | |
| Weight decay | | 0.05 | |
| SVHN augmentation | – | No horizontal flip | – |
| Other hyperparameters | | Same as Table S8 (CPolyNeXt) | |

stochastic-depth schedule as its LayerNorm baseline. All other hyperparameters (epochs, augmentation, EMA, warmup, weight initialization) follow Table S8.

# D. Computational Cost

## D.1. Inference Throughput and Memory

Table S11 compares inference throughput and peak GPU memory of our models against MetaFormer instantiations and prior polynomial networks. Our models achieve substantial memory savings compared to all baselines: CPolyNeXt and APolyNeXt use 30–45% less memory than MetaFormer at comparable scales, and roughly half the memory of prior polynomial networks. In terms of throughput, our models are slower than MetaFormer due to increased depth, but remain significantly faster than prior polynomial networks (MONet, DTTN) while achieving higher accuracy. At the Tiny scale, our models achieve over 2200 img/s, comparable to MetaFormer.

# E. Smaller-Scale Evaluation

We evaluate on three smaller benchmarks trained from scratch at native resolution (Table S12): CIFAR-10 (50K images, 32×32, 10 classes), SVHN (73K images, 32×32, 10 classes), and Tiny ImageNet (100K images, 64×64, 200 classes). CPolyNeXt-LR outperforms prior polynomial networks by large margins, including +10 points on Tiny ImageNet, confirming that our polynomial modules generalize beyond ImageNet-scale data. Configuration details are in Table S10.

*Table S11.* **Inference throughput and memory comparison.** Measured on a single NVIDIA RTX 4090 GPU with batch size 128 using `torch.compile`. * indicates activation-free models. Our models achieve substantial memory savings while remaining faster than prior polynomial networks.

| Model | Params (M) | FLOPs (G) | Memory (MB) | Throughput (img/s) |
|---|---|---|---|---|
| *MetaFormer Instantiations* | | | | |
| ConvFormer-S18 | 27 | 3.9 | 1900 | 2325 |
| ConvFormer-S36 | 40 | 7.6 | 2038 | 1369 |
| ConvFormer-M36 | 57 | 12.8 | 2892 | 975 |
| CAFormer-S18 | 26 | 4.1 | 2064 | 1807 |
| CAFormer-S36 | 39 | 8.0 | 2078 | 1019 |
| CAFormer-M36 | 56 | 13.2 | 2800 | 761 |
| *Prior Polynomial Networks* | | | | |
| MONet-T* | 10.3 | 2.8 | 1972 | 2078 |
| MONet-S* | 32.9 | 6.8 | 3434 | 1322 |
| DTTN-T* | 7.1 | 2.4 | 2694 | 1411 |
| DTTN-S* | 12.3 | 4.1 | 2676 | 1267 |
| DTTN-B* | 35.9 | 12.3 | 4640 | 701 |
| *Ours* | | | | |
| CPolyNeXt-T* | 6.4 | 1.2 | **1158** | 2358 |
| CPolyNeXt-S* | 26 | 4.8 | **1568** | 1032 |
| CPolyNeXt-B* | 40 | 8.5 | **1760** | 657 |
| CPolyNeXt-L* | 57 | 12.6 | **2036** | 474 |
| APolyNeXt-T* | 6.5 | 1.3 | **1132** | 2270 |
| APolyNeXt-S* | 26 | 5.3 | **1600** | 917 |
| APolyNeXt-B* | 41 | 9.3 | **1758** | 580 |
| APolyNeXt-L* | 57 | 13.3 | **2080** | 447 |

*Table S12.* **Smaller datasets.** CPolyNeXt-LR outperforms prior polynomial networks by large margins, including +10 points on Tiny ImageNet. Top-1 accuracy (%) trained from scratch at native resolution. ♦ indicates activation-free.

| Model | CIFAR-10 | SVHN | Tiny-IN |
|---|---|---|---|
| *Non-polynomial baselines* | | | |
| ResNet-18 (He et al., 2016) | 94.4 | 97.3 | 61.5 |
| MLP-Mixer (Tolstikhin et al., 2021) | 90.6 | 96.8 | 45.6 |
| ResMLP (Touvron et al., 2021a) | 92.3 | 97.1 | 58.9 |
| *Prior polynomial models* | | | |
| MONet-T♦ (Cheng et al., 2024) | 94.8 | 97.6 | 61.5 |
| DTTN♦ (Nie, 2025) | 95.0 | – | 63.8 |
| *Ours* | | | |
| **CPolyNeXt-LR♦** | **97.1** | **98.1** | **74.0** |

