# OpenReview forum: "Activation-Free Backbones for Image Recognition: Polynomial Alternatives within MetaFormer-Style Vision Models"
_ICML.cc/2026/Conference — ICML 2026 regular_

### Official Review · Reviewer_jJsy · 2026-03-07

**Soundness:** 3
**Presentation:** 4
**Significance:** 3
**Originality:** 3
**Overall Recommendation:** 5
**Confidence:** 4

**Summary:**

This paper investigates whether modern visual backbones can be built without standard pointwise activations or softmax attention, by replacing them with polynomial interaction modules based on Hadamard products. To this end, the paper proposes three activation-free building blocks—**PolyMLP** for channel mixing, **PolyConv** for convolutional spatial mixing, and **PolyAttn** for attention-style spatial mixing—and integrates them into MetaFormer-style backbones to obtain the proposed **CPolyNeXt** and **APolyNeXt** architectures.

Empirically, the paper evaluates these architectures on ImageNet-1K, several robustness benchmarks, and smaller image datasets, showing that the proposed polynomial backbones can match or outperform comparable activation-based MetaFormer variants as well as prior polynomial-network baselines. The paper also provides ablations on stabilization and design choices, with the overall claim that Hadamard-product-based polynomial modules can serve as effective alternatives to standard activations for both spatial and channel mixing in modern image recognition models.

**Compliance With Llm Reviewing Policy:**

Affirmed.

**Key Questions For Authors:**

1. **How much of the final performance depends on the polynomial modules themselves, and how much depends on the surrounding stabilization recipe?**
   The paper makes a strong case that PolyMLP / PolyConv / PolyAttn are effective replacements for standard activations and softmax, but the training setup also relies on a number of supporting choices, including Sigmoid-Scale, multi-input skip connections, pre-cell normalization, smaller batch size, and stronger regularization. I would like to better understand which of these are essential for competitiveness at ImageNet scale, and whether there is a minimal version of the recipe that still works well.

2. **How should the main claim be interpreted: as a statement about these specific MetaFormer-style visual backbones, or as a broader claim about the dispensability of activations in deep vision models?**
   The results are convincing in the architectures studied here, but the title-level framing is broader than the exact empirical scope. A more precise statement of intended scope would help calibrate the contribution.

3. **Do the authors have a stronger mechanistic explanation for why reintroducing activations hurts in these models?**
   Table 3 is interesting, and the result itself is surprising enough to be one of the paper’s most memorable findings. But at the moment the explanation feels mostly empirical. Even a sharper hypothesis—optimization-related, representational, or tied to the Hadamard-product construction—would strengthen the paper.

4. **How robust are the conclusions outside image classification, especially for dense prediction or transfer settings?**
   The ImageNet and robustness results are strong, but the claim is broad enough that I am curious whether the same activation-free design remains competitive in tasks where spatial precision or transfer behavior matters more directly. If the authors have any preliminary evidence here, it would help clarify the practical scope of the method.

**Limitations:**

No.

The paper does discuss some limitations, especially in **Section 5**, where it notes that the proposed models still rely on supporting design choices for stable training and that the FHE/privacy motivation is not yet fully realized because operations such as normalization remain. That said, I do not think the limitations discussion is yet fully adequate.

A stronger limitations paragraph would help by stating more explicitly:

1. **Scope of the empirical claim.**
   The current evidence is strong for ImageNet-style image classification and robustness benchmarks, but it does not yet establish that activation-free polynomial backbones are broadly competitive across other vision settings or across deep learning more generally. See **Section 4.1** and **Section 5**.

2. **Dependence on the stabilization recipe.**
   The success of the proposed models is not just about the polynomial modules in isolation; it also depends on a fairly careful surrounding recipe, including Sigmoid-Scale, skip-connection design, normalization, and training settings. This should be acknowledged more directly as a limitation on how “drop-in” the method currently is. See **Section 4.2**, **Section 4.3**, and **Section 5**.

3. **Potential societal impact.**
   I do not see a major negative societal-impact concern specific to this paper. However, if the authors want to keep the discussion complete, they could briefly note that architectural simplification or activation-free design does not by itself imply safer, fairer, or more privacy-preserving deployment, and that those properties would still require task-specific validation.

Overall, the paper is reasonably responsible, but the limitations section would be stronger with a more explicit statement of scope, recipe dependence, and the current gap between the architectural result and the broader motivational claims.

**Strengths And Weaknesses:**

**Strengths.**
The paper addresses a clear and interesting question: whether standard pointwise activations and softmax are truly necessary in modern visual backbones, or whether polynomial interactions alone can provide sufficient nonlinearity for strong image recognition performance. This is a meaningful **significance** angle because it challenges a widely used design assumption in deep learning and does so in a practically relevant setting.

A major strength is the paper’s **originality** at the architectural level. Rather than proposing a single isolated module, it develops a fairly systematic activation-free design space with three components—**PolyMLP**, **PolyConv**, and **PolyAttn**—covering channel mixing, convolutional spatial mixing, and attention-style spatial mixing. This makes the contribution broader and more coherent than a one-off block replacement.

The paper is also strong on **soundness** from an empirical perspective. It evaluates the proposed architectures across multiple model sizes on **ImageNet-1K**, includes comparisons to activation-based MetaFormer variants as well as prior polynomial-network baselines, and further reports results on robustness benchmarks such as **ImageNet-C/A/R/Sketch**. This gives the paper a solid empirical foundation and makes the main claim much more convincing than if it relied only on a narrow benchmark set.

Another strength is that the paper includes useful ablations and implementation analysis. The study of adding back activations, the role of Sigmoid-Scale, multi-input skip connections, and the depth-versus-width comparison all help support the claim that the proposed design is not simply a superficial variant but requires a distinct stabilization recipe. This improves both the paper’s empirical credibility and its practical value.

The **presentation** is generally strong. The paper is clearly structured, the method is easy to follow, and the narrative from motivation to module design to large-scale evaluation is coherent. The figures and tables are informative, and the paper does a good job of situating itself relative to prior polynomial-network work and MetaFormer-style backbones.

**Weaknesses.**
My main concern is that some of the headline framing is somewhat stronger than what the evidence strictly supports. The paper argues that activations are unnecessary, but the actual evidence is more specific: under the proposed architectural choices and stabilization recipe, Hadamard-product-based polynomial modules can replace standard activations and softmax in these visual backbone settings. That is already a strong result, but it is narrower than the broadest interpretation of the title-level claim. This affects both **presentation** and, to a lesser extent, **significance**. See **Section 1**, **Section 3**, and **Section 5**.

A second concern is that the method contribution is somewhat entangled with a fairly careful training and stabilization recipe. The paper shows that successful deep polynomial backbones rely on ingredients such as smaller batch size, stronger regularization, Sigmoid-Scale, pre-cell normalization, and skip-connection design. This does not invalidate the results, but it makes the contribution less purely about the polynomial modules themselves and more about the combined architecture-plus-training design. From a **soundness** and **originality** standpoint, it would be helpful to better isolate how much of the gain comes from the core modules versus the surrounding recipe.

The theoretical understanding is also limited. While the paper clearly defines the polynomial modules and notes their structural properties, it does not provide a strong theoretical explanation for why these activation-free backbones train stably at scale, why reintroducing activations can hurt, or why the proposed models appear relatively robust. As a result, the work is primarily an empirical architecture paper rather than one that offers deep conceptual understanding of the phenomenon. This limits the paper’s **soundness** only mildly, but more importantly tempers its **significance** and long-term explanatory value.

Finally, some of the broader motivation around privacy-preserving or FHE-friendly inference remains preliminary. The paper correctly notes that removing nonlinear activations may be useful in such contexts, but it also acknowledges that operations such as LayerNorm still remain obstacles. At present, this part reads more like a potential future direction than a demonstrated practical implication.

Overall, I view the paper as a strong empirical architecture contribution with clear novelty and substantial experimental support, but with somewhat overstated framing and limited theoretical explanation relative to the ambition of its central claim.

---

> ### Author Rebuttal · Authors · 2026-03-31
>
> We thank Reviewer jJsy for their exceptionally helpful review. We address each question with new evidence and will include all results in the revised version.
>
> ---
>
> > ### **Q1: Does the final performance depend more on the modules or the recipe?**
>
> The modules. The key finding: optimizer choice is where model-specificity lives, while regularization transfers well. CPolyNeXt-S achieves 83.7% with ConvFormer's optimizer ($-$0.2), while ConvFormer-S18 collapses to ~6% under ours. Regularization swaps produce only minor effects in both directions (ConvFormer +0.1, CAFormer $-$0.3 under our settings). This demonstrates that recipe differences reflect genuine differences in optimization landscape, not a universal advantage for either recipe. Please see [Rev 7CrJ Q2](https://openreview.net/forum?id=FR1XRg47fL&noteId=uiL7r6Pez8) for the full cross-recipe analysis.
>
> ---
>
> > ### **Q2: Scope and claim calibration.**
>
> Inspired by the reviewer’s comment, we will update the title to "Activation-Free Backbones for Image Recognition: Polynomial Alternatives within MetaFormer-Style Vision Models" and add qualifiers: the abstract will read "we demonstrate they are not required within MetaFormer-style vision backbones," and Contribution #2 will specify "in MetaFormer-style vision backbones." We will also expand the conclusion to include our segmentation results.
>
> That said, our new results have substantially widened the empirical scope since submission:
>
> *Segmentation.* We evaluated on ADE20K [Zhou et al., IJCV 2019] semantic segmentation using UperNet [Xiao et al., ECCV 2018] (160K iters, standard ConvNeXt [Liu et al., CVPR 2022] recipe, no tuning). MetaFormer baseline results are from Yu et al. [TPAMI 2024]:
>
> | Model | Params (M) | MACs (G) | mIoU |
> |-------|-----------|----------|------|
> | ConvFormer-S18 | 54 | 925 | 48.6 |
> | CAFormer-S18 | 54 | 1024 | 48.9 |
> | **CPolyNeXt-S** | **54** | **941** | **50.6** |
> | **APolyNeXt-S** | **55** | **1121** | **49.9** |
>
> CPolyNeXt-S surpasses both baselines (+2.0 over ConvFormer, +1.7 over CAFormer) at matched parameters and matches ConvFormer-S36 (50.7 mIoU, 67M params, 1003G MACs) at 54M params and 942G MACs. Segmentation gains exceed our classification margins, suggesting polynomial backbones transfer particularly well. APolyNeXt-S required per-cell gradient clipping (identity on forward, clips gradient norm at cell boundaries during backpropagation; norm=5); the conv variant needed no modifications.
>
> *Module drop-in.* We replaced the spatial mixers in CAFormer-S18 with PolyConv and PolyAttn respectively, keeping CAFormer's architecture, training recipe, and matching FLOPs/parameters. Both improve ImageNet accuracy by +0.1 to +0.3, confirming module-level generality beyond our architecture.
>
> ---
>
> > ### **Q3: Why do activations hurt?**
>
> In the backward pass of the block $(W_ax) * (W_bx)$, both branches depend on $x$ and thus there is a mutual coupling of the two. Adding an activation like GELU breaks this mutual coupling: GELU's near-zero derivative for negative inputs creates dead zones where gradient flow is suppressed. This explains the ordering in Table 3, where $\sigma$ denotes GELU ($\Delta$ values are accuracy differences from the 80.2% baseline):
>
> - **One branch** ($\sigma(W_ax) * (W_bx)$, $\Delta = -0.4$): only one gradient path is partially blocked.
> - **Both branches** ($\sigma(W_ax) * \sigma(W_bx)$, $\Delta = -0.6$): both paths are partially blocked independently.
> - **After product** ($\sigma((W_ax) * (W_bx))$, $\Delta = -1.0$): a single activation gates the combined signal, blocking gradients to both projections simultaneously for any input where the product is negative, explaining the largest degradation.
>
> Replacing multiplication with addition ($-$22.3) removes the mutual gradient coupling entirely, confirming that this coupling is the essential source of expressive nonlinearity. We will incorporate this analysis into the Discussion section of the revised paper.
>
> ---
>
> > ### **Q4: Dense prediction and transfer.**
>
> Please see the segmentation results under Q2 above. CPolyNeXt-S achieves 50.6 mIoU on ADE20K, surpassing both ConvFormer-S18 (+2.0) and CAFormer-S18 (+1.7), using an ImageNet-pretrained backbone fine-tuned with UperNet. Please also see [Rev yoQg Q2](https://openreview.net/forum?id=FR1XRg47fL&noteId=EQHVtvi2xL) for fully polynomial variants (replacing LayerNorm with polynomial-compatible BatchNorm) achieving 82.7% on ImageNet, extending the activation-free paradigm to normalization-free inference.
>
> ---
>
> > ### **Limitations.**
>
> We will add the following sentence in the impact: Note that our activation-free design does not by itself imply safer, fairer, or more privacy-preserving deployment, and that those properties require task-specific validation, which is left for future work.
>
> ---
>
> We are grateful for the reviewer's thorough and constructive feedback. Please do not hesitate to ask if there are any additional questions.

---

> > ### Author Rebuttal · Reviewer_jJsy · 2026-04-01
> >
> > Thank you for the helpful rebuttal. My main concerns are resolved. The scope is now better calibrated, and the new evidence is meaningful: the added segmentation results broaden the empirical support, the module drop-in results better isolate the contribution of the polynomial modules from the surrounding recipe, and the added explanation of why activations hurt strengthens the interpretation of the ablation results. Together with the strong experimental evidence already in the paper, this increases my confidence in the contribution, so I am raising my score.

---

> > > ### Author Response · Authors · 2026-04-05
> > >
> > > We thank Reviewer jJsy deeply for their generous reassessment and for the detailed feedback that substantially improved our work. The reviewer's questions on module-vs-recipe isolation, scope calibration, mechanistic explanation of why activations hurt, and dense prediction robustness led to new segmentation results (e.g., module drop-in experiments, and a gradient coupling analysis) that we will incorporate into the revised paper. We are grateful for the engagement.

---

### Official Review · Reviewer_7CrJ · 2026-03-12

**Soundness:** 3
**Presentation:** 3
**Significance:** 3
**Originality:** 3
**Overall Recommendation:** 4
**Confidence:** 1

**Summary:**

This paper introduces PolyNeXt, a family of activation-free vision backbones that replaces standard nonlinearities (ReLU, Softmax) with polynomial operations (Hadamard products).

**Compliance With Llm Reviewing Policy:**

Affirmed.

**Final Justification:**

Thanks for the rebuttal. My concerns are completely resolved. i will keep my rating.

**Key Questions For Authors:**

1. Can you provide a direct comparison of actual inference throughput against MetaFormer baselines to clarify the practical cost of the depth-over-width design?

2. How catastrophic is the failure if standard vision training recipes are applied?

**Limitations:**

yes

**Strengths And Weaknesses:**

Strengths:The paper challenges a deeply entrenched assumption in modern deep learning: that pointwise activation functions (e.g., ReLU, GELU) and exponential softmax are strictly necessary for competitive vision backbones. By completely replacing these with polynomial operations (Hadamard products) across all three core primitives (MLP, Convolution, and Attention) while maintaining standard input-output interfaces, the authors present a highly original, and theoretically refreshing perspective on architectural design.

Weaknesses:
1. PolyNeXt uses a substantially different training regime from its MetaFormer baselines, incorporating a smaller batch size, progressive dropout, an EMA for the last 200 epochs, and custom-tuned stochastic depth.

2.  While the paper frequently frames the polynomial networks as computationally efficient based on theoretical FLOPs, the actual wall-clock throughput penalty is severe. Halving the inference throughput is a massive practical cost in real-world deployments, and this trade-off deserves much more upfront emphasis in the main text rather than being relegated to the appendix.

3.  FHE motivation is overstated: The paper repeatedly invokes Fully Homomorphic Encryption compatibility as a motivation, but LayerNorm (division + square root) remains in every module.

---

> ### Author Rebuttal · Authors · 2026-03-31
>
> We thank Reviewer 7CrJ for their review and for highlighting the originality of replacing all three core primitives. We address each concern below and will incorporate the new results into the revised paper.
>
> ---
>
> > ### **Q1: Inference throughput.**
>
> As reported in Appendix Table S9, our depth-over-width design incurs a general throughput reduction compared to MetaFormer across scales. However, two factors provide important context. First, our models use significantly less peak GPU VRAM during inference at matched batch sizes, which is valuable for deployment under hardware constraints and particularly for FHE inference where encrypted computation is memory-intensive. Second, compared to prior polynomial networks at similar accuracy, our models achieve comparable or better throughput with far less memory:
>
> | Model | Acc | Tput (img/s) | Mem (MB) |
> |-------|-----|-------------|----------|
> | DTTN-T | 77.9% | 1411 | 2694 |
> | MONet-T | 77.0% | 2078 | 1972 |
> | **CPolyNeXt-T** | **80.2%** | **2358** | **1158** |
> | DTTN-B | 82.4% | 701 | 4640 |
> | **CPolyNeXt-B** | **84.7%** | **657** | **1760** |
>
> At the Tiny scale, CPolyNeXt-T is faster, more accurate, and uses 41--57% less memory than both prior PNs. At larger scales, CPolyNeXt-B matches DTTN-B in throughput while being 2.3 points more accurate and using 62% less memory. The throughput gap vs. MetaFormer stems from sequential depth: more layers means more kernel launches and less GPU parallelism per layer, even at matched FLOPs. Since our current implementation launches a separate kernel for each narrow operation, this overhead is substantial but not fundamental; fusing consecutive operations into fewer kernels would reduce launch overhead and keep intermediates in on-chip SRAM, disproportionately benefiting our deeper architecture. We will add a quantitative summary of the throughput-memory trade-off to the main results section in the revised paper, so readers do not need to consult the appendix for this information.
>
> ---
>
> > ### **Q2: Failure under standard training recipes.**
>
> To address the question, we conducted cross-recipe experiments at S (Small, 26M params) scale. To ensure fair comparison, we retrained ConvFormer-S18 and CAFormer-S18 in their codebase under the same speedups (torch.compile, AMP bf16) and verified accuracy is ~$-$0.1 of published numbers (ConvFormer-S18: 82.9%, CAFormer-S18: 83.5%). All cross-recipe runs use EMA from epoch 200, reporting the best of EMA and standard checkpoints.
>
> | Configuration | $\Delta$ from baseline |
> |--------------|----------------|
> | *Our models w/ ConvFormer settings* | *(baselines: CPolyNeXt-S 83.9%, APolyNeXt-S 84.3%)* |
> | CPolyNeXt-S w/ ConvFormer optimizer | $-$0.2 |
> | APolyNeXt-S w/ ConvFormer optimizer | $-$0.4 |
> | *ConvFormer-S18 w/ our settings* | *(baseline: 82.9%)* |
> | ConvFormer-S18 w/ our optimizer (w/ or w/o our reg) | collapse (~6%) |
> | ConvFormer-S18 w/ our regularization | +0.1 |
> | *CAFormer-S18 w/ our settings* | *(baseline: 83.5%)* |
> | CAFormer-S18 w/ our optimizer | $-$0.4 |
> | CAFormer-S18 w/ our regularization | $-$0.3 |
> | CAFormer-S18 w/ our full recipe | $-$0.3 |
>
> The results reveal that optimizer choice is where model-specificity lives, while regularization transfers well in both directions. ConvFormer under our regularization is essentially unchanged (+0.1), and CAFormer under our full recipe drops only 0.3; but ConvFormer collapses under our optimizer. On the other hand, both CPolyNeXt-S ($-$0.2) and APolyNeXt-S ($-$0.4) are nearly unaffected by ConvFormer's optimizer. Notably, these drops are comparable to what MetaFormer models themselves experience under recipe swaps (CAFormer $-$0.3 to $-$0.4), demonstrating that our polynomial variants are no more recipe-sensitive than standard architectures.
>
> Although all competitive architectures involve model-specific optimizer tuning, this evidence shows that the polynomial modules themselves do not introduce unusual sensitivity.
>
> ---
>
> > ### **Q3: FHE motivation.**
>
> We believe the FHE motivation is now substantiated by concrete results. Inspired by this question, we train fully polynomial variants achieving 78.0 - 82.7% on ImageNet, a 5.0-point advance over the best prior fully polynomial model (MONet-T, 77.7%). By replacing LayerNorm with polynomial-compatible BatchNorm, the entire inference pass involves only additions and multiplications, directly addressing the LayerNorm barrier the reviewer identified. Please see [Rev yoQg Q2](https://openreview.net/forum?id=FR1XRg47fL&noteId=EQHVtvi2xL) for the full results.
>
> ---
>
> We thank the reviewer for their time and engagement. Please let us know if any questions remain.

---

> > ### Author Rebuttal · Reviewer_7CrJ · 2026-04-02
> >
> > Thanks for the additional explanation and experiments. My concerns are resolved.

---

> > > ### Author Response · Authors · 2026-04-05
> > >
> > > We sincerely thank Reviewer 7CrJ for confirming that their concerns are resolved. The reviewer's questions on inference throughput, training recipe sensitivity, and FHE motivation led to new cross-recipe experiments and fully polynomial model variants that strengthen the paper. We will incorporate all new results and move the throughput-memory analysis to the main text as committed.

---

### Official Review · Reviewer_yoQg · 2026-03-14

**Soundness:** 3
**Presentation:** 3
**Significance:** 3
**Originality:** 3
**Overall Recommendation:** 5
**Confidence:** 4

**Summary:**

This paper investigates whether traditional pointwise activations (e.g., ReLU, GELU) and exponential softmax are strictly necessary for modern vision backbones. The authors propose PolyNeXt, a hierarchical architecture that replaces these primitives with polynomial alternatives, PolyMLP, PolyConv, and PolyAttn, which utilize Hadamard products to generate nonlinearity. To overcome the stability challenges of training deep polynomial networks, the authors introduce a specialized stabilization recipe featuring Sigmoid-Scale residual gating, multi-input skip connections, and a depth-over-width design philosophy. Empirical results on ImageNet-1K demonstrate that these activation-free models match or outperform standard counterparts like MetaFormer while offering improved robustness to out-of-distribution data and significantly lower memory consumption.

**Compliance With Llm Reviewing Policy:**

Affirmed.

**Final Justification:**

Thank you for the rebuttal. I will raise my score.

**Key Questions For Authors:**

See weaknesses

**Limitations:**

See weaknesses

**Strengths And Weaknesses:**

# Strengths

1. The authors introduce a creative stabilization recipe, specifically Sigmoid-Scale residual gating and multi-input skip connections, to solve the long-standing instability issues in deep polynomial networks. This allows them to train networks up to 200 layers deep, a major shift from prior polynomial architectures that were restricted to shallower, wider designs to avoid numerical explosion.

2. The paper is methodologically rigorous, demonstrating that PolyNeXt matches or exceeds standard activation-based counterparts (MetaFormer) on ImageNet-1K. Furthermore, it provides strong evidence of superior out-of-distribution robustness on benchmarks like ImageNet-A/R/Sketch, often outperforming models with significantly more parameters.

3. The work offers high practical utility by achieving 30-45% memory savings compared to standard MetaFormer models. Additionally, by removing non-polynomial activations, it provides a foundational step toward neural networks compatible with Fully Homomorphic Encryption (FHE), which is crucial for privacy-preserving machine learning.

# Weaknesses
1. While the "depth-over-width" strategy is effective for accuracy, it incurs a significant cost in inference throughput. Comparative data in Table S9 shows that PolyNeXt models are consistently slower (in images per second) than their MetaFormer baselines, which may hinder their adoption in real-time applications.

2. Despite the motivation to create FHE-amenable models, the architecture still relies on LayerNorm, which involves division and square root operations that are non-polynomial. Consequently, the paper moves the needle forward but does not yet deliver a fully activation-free model that can be used for end-to-end FHE inference without further modifications.

---

> ### Author Rebuttal · Authors · 2026-03-31
>
> We thank Reviewer yoQg for their thorough and positive assessment, and for recognizing the memory efficiency and FHE potential of our approach. We address the questions below and we will clarify them in the revised version.
>
> ---
>
> > ### **Q1: Throughput overhead.**
>
> We agree that the throughput cost of our depth-over-width design is a meaningful practical consideration. The gap stems from sequential depth: more layers means more kernel launches and less GPU parallelism per layer, even at matched FLOPs. As the reviewer notes, our models offer substantial memory savings, which partially offsets this cost in memory-constrained deployments. This is particularly relevant for FHE, where encrypted computation is memory-intensive. Compared to prior polynomial nets, our models dominate the accuracy-throughput-memory Pareto front (see [Rev 7CrJ Q1](https://openreview.net/forum?id=FR1XRg47fL&noteId=uiL7r6Pez8)). We also note our implementation uses no custom CUDA kernels; fusing sequences of narrow operations into fewer kernels (reducing launch overhead and roundtrips to high-bandwidth memory) would disproportionately benefit our deeper architecture and narrow the MetaFormer gap.
>
> ---
>
> > ### **Q2: FHE and LayerNorm.**
>
> Inspired by the reviewer's question, we train fully polynomial variants that directly replace LayerNorm. We replace all LayerNorm with polynomial-compatible BatchNorm. In a standard BCHW tensor, conventional BatchNorm normalizes over the batch and spatial dimensions (B, H, W) with affine parameters over channels (C). Our variant instead normalizes over the first two dimensions (B and C, i.e., batch and channels), with affine parameters over both channels and spatial positions, and uses running statistics at inference. For the attention L1 normalization, we use the row-sum averaged over the batch as a running statistic with learnable affine parameters. The entire inference pass thus involves only additions and multiplications. The results are below:
>
> | Model | Params | FLOPs | LN ver. | Poly BN | Drop |
> |-------|--------|-------|---------|---------|------|
> | *Prior polynomial nets* | | | | | |
> | MONet-T (BN) | 10.3M | 2.8G | 77.0% | 77.7% | +0.7 |
> | DTTN-S (no norm) | 12.3M | 4.1G | 79.4% | 77.2% | $-$2.2 |
> | *Ours* | | | | | |
> | CPolyNeXt-T | 6.4M | 1.2G | 80.2% | 78.3% | $-$1.9 |
> | APolyNeXt-T | 6.5M | 1.3G | 80.9% | 78.0% | $-$2.9 |
> | CPolyNeXt-S | 26M | 4.8G | 83.9% | 82.7% | $-$1.2 |
>
> Our CPolyNeXt-S BN variant at 82.7% represents a 5.0-point improvement over the best prior fully polynomial result (MONet-T, 77.7%). MONet's BatchNorm variant gains only +0.7 on its smallest model and remains far below our accuracy. Even our smallest BN model (CPolyNeXt-T, 78.3%) exceeds DTTN-S despite having roughly half the parameters and FLOPs.
>
> Notably, CPolyNeXt-S BN (82.7%, 26M params) surpasses ConvNeXt-T (82.1%, 29M params), a strong activation-based baseline, at fewer parameters, and approaches ConvFormer-S18 (83.0%) within 0.3 points. This demonstrates that fully polynomial, FHE-compatible models need not sacrifice competitiveness with mainstream architectures.
>
> The APolyNeXt-T gap (2.9 pts vs 1.9 for CPolyNeXt-T) is larger due to the additional approximation needed for the L1 attention normalization using running statistics. The CPolyNeXt-S result (82.7%, only 1.2 points below LayerNorm) shows this cost diminishes at scale.
>
> ---
>
> We appreciate the reviewer's time and thoughtful feedback. Please let us know if there are any further questions.

---

> > ### Author Rebuttal · Reviewer_yoQg · 2026-04-03
> >
> > Thank you for the rebuttal. I will raise my score.

---

> > > ### Author Response · Authors · 2026-04-05
> > >
> > > We sincerely thank Reviewer yoQg for engaging with our rebuttal and for raising their score. The reviewer's questions on throughput overhead and FHE compatibility motivated our fully polynomial BatchNorm variants, which we believe are a meaningful contribution beyond the original submission. We will incorporate all new results into the revised paper.

---

### Official Review · Reviewer_mhNt · 2026-03-20

**Soundness:** 3
**Presentation:** 3
**Significance:** 3
**Originality:** 3
**Overall Recommendation:** 4
**Confidence:** 2

**Summary:**

This paper challenges the assumption that standard nonlinearities are necessary in vision models. It introduces activation-free polynomial alternatives for MLPs, convolutions, and attention, using Hadamard products to model nonlinearity within a unified framework. The approach matches or outperforms conventional models while being more efficient than prior polynomial networks.

**Compliance With Llm Reviewing Policy:**

Affirmed.

**Final Justification:**

The rebuttal addressed my questions. I keep my initial positive recommendation.

**Key Questions For Authors:**

See Weakness

**Limitations:**

Yes

**Strengths And Weaknesses:**

### Strengths

- **Strong Empirical Performance:** The paper demonstrates strong empirical results. The proposed models significantly outperform prior polynomial networks and match or slightly exceed standard, activation-based architectures across multiple model scales.
- **Thorough Ablation Studies:** The authors provide detailed ablation studies that isolate the contribution of each building block.
- **Transparency:** The paper is transparent about its drawbacks, dedicating a clear section to its limitations.

### Weaknesses

- **"Activation-Free" Claim:** A conceptual weakness is the reliance on the "Sigmoid-Scale" mechanism to achieve training stability. While the authors argue that this sigmoid function is applied to a learnable scalar that can be absorbed into the weights at inference, the network's ability to train without collapsing might still rely on the bounding effects of a non-linear activation function. This somewhat undermines the paper's core "activation-free" claim.
- **Highly Specialized Training Recipe:** As the authors acknowledge, the models require a highly specialized and delicate training recipe. This means these polynomial modules cannot be easily "dropped in" to standard training pipelines.
- **Throughput Overhead:** The authors' depth-over-width design is necessary to increase the polynomial degree and maintain stability, but it also comes with the cost of throughput overhead, even when parameter counts and FLOPs are matched.

---

> ### Author Rebuttal · Authors · 2026-03-31
>
> We thank Reviewer mhNt for their review and for recognizing the strong empirical performance, thorough ablations, and transparency of our work. We address the concerns below and we will include all the related results in the revised version.
>
> ---
>
> > ### **Q1: Sigmoid-Scale and training stability.**
>
> The reviewer correctly identifies that $\sigma(\lambda)$ can be absorbed at inference; the question is whether the bounding effect of sigmoid is essential during training. Our experimentation indicates that sigmoid is not strictly essential; the initialization geometry is the primary factor, though sigmoid does provide a secondary optimization benefit.
>
> Sigmoid-Scale initializes residual contributions to decrease geometrically with depth within each cell ($\sigma(\lambda_0)=0.50$, $\sigma(\lambda_1)\approx0.38$, $\sigma(\lambda_2)\approx0.27$, ...; Appendix A.2). This creates a depth hierarchy where shallower sublayers establish representations before deeper sublayers contribute substantially. The sigmoid maps unconstrained logits to (0,1), modulating gradient magnitude accordingly; deeper sublayers with smaller $\sigma(\lambda)$ naturally receive smaller gradient updates, stabilizing early training. This role is analogous to learning rate warmup or LayerScale, i.e., an optimization aid, not a source of nonlinearity in the learned function.
>
> To directly test this, we train CPolyNeXt-T, replacing $\sigma(\lambda)$ with a learnable scalar initialized to the same value. The model achieves 79.7%, only 0.5 below our 80.2% baseline. For context, replacing Sigmoid-Scale with LayerScale (a learnable per-channel vector) costs 0.8 points even at small initialization ($10^{-6}$) and causes training collapse at standard initialization ($-$12.8; Table S5). The result at $-$0.5 confirms that the initialization hierarchy is the primary mechanism, not the sigmoid nonlinearity. The remaining 0.5-point gap reflects a secondary optimization benefit: when parameterized through sigmoid, the effective update to each scale is modulated by $\sigma'(\lambda) = \sigma(\lambda)(1-\sigma(\lambda))$, which is smaller for deeper sublayers (those initialized with smaller $\sigma(\lambda)$). This naturally ties update magnitude to current contribution size, so deeper sublayers adjust more slowly. A free learnable scalar lacks this property, receiving updates of equal magnitude regardless of its current value. We will include this ablation in the revised paper.
>
> ---
>
> > ### **Q2: Training recipe.**
>
> Our cross-recipe experiments (training each model with the other's optimizer and regularization settings) show the conv variant is nearly recipe-agnostic: CPolyNeXt-S achieves 83.7% with ConvFormer's published optimizer settings, only 0.2 points below our original 83.9%. Optimizer sensitivity is model-specific; ConvFormer-S18 collapses under our optimizer, while regularization changes are benign in both directions. Please see [Rev 7CrJ Q2](https://openreview.net/forum?id=FR1XRg47fL&noteId=uiL7r6Pez8) for the full cross-recipe analysis. We will include the full ablation in the camera-ready version.
>
> ---
>
> > ### **Q3: Throughput.**
>
> As discussed in Appendix Table S9, our depth-over-width design incurs a throughput reduction compared to MetaFormer across scales. The gap stems from sequential depth: more layers means more kernel launches and less GPU parallelism per layer, even at matched FLOPs. However, our models use less peak GPU VRAM during inference (at matched batch sizes), which is important for memory-constrained deployment and particularly for FHE. Compared to prior polynomial networks, our models achieve higher accuracy with comparable or better throughput and substantially less memory. Our implementation uses no custom CUDA kernels; fusing sequences of narrow operations into fewer kernels (reducing launch overhead and keeping intermediates in on-chip SRAM) would disproportionately benefit our architecture and narrow the gap. Please see [Rev 7CrJ Q1](https://openreview.net/forum?id=FR1XRg47fL&noteId=uiL7r6Pez8) for the throughput comparison table.
>
> ---
>
> We are thankful for your time and effort. Please let us know if there are any additional questions.

---

> > ### Author Rebuttal · Reviewer_mhNt · 2026-04-06
> >
> > Thanks for the response. I'll keep my positive score.

---

> > > ### Author Response · Authors · 2026-04-07
> > >
> > > We thank Reviewer mhNt for confirming that their concerns are resolved. The reviewer's questions on training stability, recipe sensitivity, and throughput overhead led to a new learnable scalar ablation and cross-recipe experiments that we will incorporate into the revised paper. We appreciate the engagement.

---

### Decision · Program_Chairs · 2026-04-30

**Decision:**

Accept (regular)

**Comment:**

This paper proposes a novel vision architecture that alleviates the need for traditional pointwise activation functions such as ReLU - and instead uses polynomial modules - where nonlinearities are replaced by pointwise multiplications (Hadamard product). Developing neural networks that are very intricate polynomials has important applications to computations on encrypted data. The proposed architecture is competitive compared to state-of-the-art small networks like MetaFormer, and clearly outperform previous models of this kind (by ~3% on ImageNet). The results presented in this paper are strong, and all reviewers acknowledge the very convincing empirical validation.
While very well executed, the proposed architecture still requires a few "classical" layers, and there remains some doubt about the relative improvement brought by more robust training recipes, versus better architectural choices.
Nonetheless, this is a very solid piece of work, and for all the above reasons I recommend this paper for acceptance.